# PARAMETER-FREE VARIANCE REDUCED ZEROTH-ORDER OPTIMIZATION FOR NONCONVEX PROBLEMS

## ABSTRACT

Zeroth-order optimization has become a vital tool for solving black-box learning problems where explicit gradients are unavailable. However, standard zeroth-order methods typically require careful tuning of algorithmic parameters such as the smoothing parameter and step size, which limits their practicality. In this paper, we propose PF-VRZO(Parameter free variance reduced zeroth-order methods), a novel parameter-free variance-reduced zeroth-order optimization framework for nonconvex finite-sum problems. Our method only requires minimal input information—problem dimension $d$ and sample size $n$—and adaptively adjusts the smoothing and step size parameters during the optimization process. We develop two algorithmic variants based on coordinate-wise and random-direction gradient estimators, respectively. We establish non-asymptotic convergence guarantees showing that PF-VRZO achieves function query complexity of $\widetilde{\mathcal{O}}(d\sqrt{n}\epsilon^{-2})$ for finding stationary points. Additionally, we conduct experiments on nonconvex phase retrieval and distributional robust optimization to validate the effectiveness of our method. To the best of our knowledge, PF-VRZO is the first parameter-free zeroth-order algorithm that incorporates variance reduction techniques tailored specifically for nonconvex optimization problems.

## 1 INTRODUCTION

In the paper, we consider solving the following stochastic nonconvex finite-sum optimization problems. $f : \mathbb{R}^d \to \mathbb{R}$

$$\underset{x \in \mathbb{R}^d}{\text{minimize}} \quad f(x) = \frac{1}{n} \sum_{i=1}^{n} f_i(x) \tag{1}$$

where $f(x)$ and each $f_i(x)$ are both smooth and possibly nonconvex functions, which captures the standard empirical risk minimization problems in machine learning.

In many important applications, computing explicit gradients is either computationally expensive or infeasible, and only function evaluations are available. Such applications include black-box adversarial attacks on deep neural networks (DNNs) (Papernot et al., 2017; Chen et al., 2017), reinforcement learning (Malik et al., 2018; Kumar et al., 2020), and fine-tuning large-scale models (Malladi et al., 2023). Zeroth-order optimization has thus emerged as a fundamental research direction (Ghadimi & Lan, 2013; Liu et al., 2018b;a; Ji et al., 2019; Lian et al., 2016; Gu et al., 2018), serving as a prototype framework for a wide range of these gradient-free learning tasks. However, a common drawback of standard zeroth-order methods is the introduction of an additional smoothing parameter $\mu$. As illustrated in Figure 1, improper tuning of this parameter in practice can lead to suboptimal performance, or even cause the algorithm to diverge.

On the other hand, recent years have seen a growing body of work on parameter-free algorithms(Ivgi et al., 2023; Kreisler et al., 2024; Orabona & Tommasi, 2017; Chen et al., 2022; Defazio & Mishchenko, 2023), particularly in the first-order setting. Several studies have demonstrated that such methods can achieve convergence rates comparable to those of parameter-dependent algorithms, even under nonconvex conditions. We define a parameter-free method as one that does not require prior knowledge of problem-specific parameters such as the smoothness constant $L$, the target accuracy $\epsilon$, or the total number of iterations $T$. This is particularly important in practical applications, where such information is typically unavailable—for example, it is often unclear how many iterations are needed,

or how small the gradient or objective value should be for the model to be considered good enough. Our expectation for a parameter-free algorithm is that it can be executed with only minimal and readily available inputs, such as the sample size $n$ and the problem dimension $d$, and run continuously until the model reaches a desirable state—such as sufficiently high test accuracy or low generalization error.

Although recent works have achieved satisfactory theoretical progress for first-order algorithms, research on zeroth-order counterparts remains quite limited. It was not until recently that Ren & Luo (2025) proposed the first parameter-free zeroth-order algorithm. Unfortunately, the theoretical guarantee of this method holds only under the assumption that the objective function $f(x)$ is convex and defined over a bounded domain. As acknowledged by the authors, extending this result to the nonconvex setting is nontrivial.

> Q1. When zeroth-order optimization meets adaptive methods, how does the error introduced by inexact gradient estimation accumulate throughout the optimization process, and is such error controllable? Can we design an adaptive algorithm that keeps this error within an acceptable range?

A1:Based on our results, after $T$ iterations, the accumulated error is approximately $\mathcal{O}[\frac{1}{T}(\sum_{t=0}^{T-1}\mu_t^2 + \sum_{t=0}^{T-1}\mu_t) + \frac{1}{\sqrt{T}}\left(n^{\frac{5}{4}}\sum_{t=0}^{T-1}\mu_t^2 + n^{\frac{1}{2}}\sqrt{\sum_{t=0}^{T-1}2\mu_t^2}\right)]$. To ensure convergence, it is crucial that condition $\sum_{t=0}^{T-1}\mu_t \leq \mathcal{O}(\sqrt{T}), \quad \sum_{t=0}^{T-1}\mu_t^2 \leq \mathcal{O}(1)$ holds; otherwise, the algorithm may diverge. This observation reveals that the error grows with $T$. A natural idea, therefore, is to let the smoothing parameter $\mu$ depend on $T$, which would directly guarantee $\sum_{t=0}^{T-1}\mu_t \leq \mathcal{O}(\sqrt{T}), \quad \sum_{t=0}^{T-1}\mu_t^2 \leq \mathcal{O}(1)$. However, this approach conflicts with our goal of designing a parameter-free algorithm, since the required number of iterations $T$ is unknown in advance. To overcome this difficulty while preserving the parameter-free property, we introduce a smart adaptive parameter $\mu_t = \frac{1}{(t+1)\sqrt{nd}}$, which evolves automatically during the optimization process to enforce $\sum_{t=0}^{T-1}\mu_t \leq \mathcal{O}(\sqrt{T}), \quad \sum_{t=0}^{T-1}\mu_t^2 \leq \mathcal{O}(1)$, without the need for any manually tuned parameters.

> Q2:Would the smoothing parameters $\mu$ that vary with $t$, as discussed above, conflict with the proof techniques of variance reduction methods? Taking the Spider estimator $v^t = \nabla f_{i_t}(x^t) - \nabla f_{i_t}(x^{t-1}) + v^{t-1}$ as an example, can we directly replace the terms $\nabla f_{i_t}(x^t)$ and $\nabla f_{i_t}(x^{t-1})$ in the Spider estimator with the zeroth-order estimators $\bar{\nabla}_{\mu_1}f_{i_t}(x_t)$ and $\bar{\nabla}_{\mu_2}f_{i_t}(x_{t-1})$? Moreover, can these two zeroth-order estimators be computed directly using the adaptive smoothing parameter $\mu_1 = \mu_2 = \mu_t = \frac{1}{(t+1)\sqrt{nd}}$?

A2: We found that directly using the smoothing parameters mentioned above in gradient estimation within variance-reduced methods does not work. This is because the convergence proofs for variance reduction often rely on the recursive relation $\mathbb{E}\|v_t - \bar{\nabla}_{\mu_t}f(x_t)\|^2 \leq \mathbb{E}\|v_{t-1} - \bar{\nabla}_{\mu_{t-1}}f(x_{t-1})\|^2 + $ (additional terms) holding exactly. To ensure this recursive relation, $\mathbb{E}[\bar{\nabla}_{\mu_1}f_{i_t}(x_t) - \bar{\nabla}_{\mu_2}f_{i_t}(x_{t-1})] = \bar{\nabla}_{\mu_t}f(x_t) - \bar{\nabla}_{\mu_{t-1}}f(x_{t-1})$ is required. Therefore, simply setting $\mu_1 = \mu_2 = \frac{1}{(t+1)\sqrt{nd}}$ does not suffice; a slight modification is needed, where we set $\mu_1 = \frac{1}{(t+1)\sqrt{nd}}$ and $\mu_2 = \frac{1}{(t)\sqrt{nd}}$.

By addressing the aforementioned challenges, this paper introduces the Parameter-Free Variance-Reduced Zeroth-Order (PF-VRZO) method, a novel approach that combines the strengths of adaptive algorithms with variance reduction techniques. Our method eliminates the need for manual parameter tuning by adaptively adjusting the smoothing parameter and step size during the optimization process. Specifically, we propose two variants of PF-VRZO: one based on coordinate-wise gradient estimators and another leveraging random-direction estimators.

Table 1: Convergence property comparison of the PF-VRZO algorithms for finding an $\epsilon$-stationary point.**C**, **NC**, **S**, and **NS** denote convex, nonconvex, smooth, and non-smooth settings, respectively. **VR** indicates whether the method is compatible with variance reduction techniques. $\sigma$ denotes an upper bound on the variance of stochastic gradients, and $D_x$ represents the diameter of the domain. The term "complexity" refers to function query complexity. Here, $\eta_t$ denotes the step size, $\mu_t$ the smoothing parameter, $c$ a generic constant, and $T$ the total number of iteration rounds. $g_t$ refers to the zeroth-order gradient estimator, while $v_t$ denotes the SPIDER estimator.*denotes deterministic case

| Method | Problem | VR? | Param-free? | Complexity | $\eta_t$ | $\mu_t$ |
|---|---|---|---|---|---|---|
| POEM (Ren & Luo) | C-NS | ✗ | ✓ | $\tilde{\mathcal{O}}(d\epsilon^{-2}D_x)$ | $\frac{\max_t\{\|x_t-x_0\|\}}{\sum_{s=0}^t \|g_t\|^2}$ | $\frac{d\max_t\{\|x_t-x_0\|\}}{t+1}$ |
| JAGUAR (Veprikov et al.) | NC-S | ✓ | ✗ | $^*\mathcal{O}(d\epsilon^{-2})$ | $\frac{1}{dL}$ | $\mathcal{O}(\frac{\epsilon}{\sqrt{d}L})$ |
| ZO-SGD (Ghadimi & Lan) | NC-S | ✗ | ✗ | $\mathcal{O}(\sigma^2\epsilon^{-4})$ | $o(\frac{1}{\sqrt{d}}\min\{\frac{1}{L\sqrt{d}},\frac{c}{\sigma\sqrt{d}}\})$ | $o(\frac{c}{d\sqrt{T}})$ |
| ZO-SPIDER-rand (Fang et al.) | NC-S | ✓ | ✗ | $\mathcal{O}(d\sqrt{n}\epsilon^{-2})$ | $\min\{\frac{c\epsilon}{L\|v_t\|},\frac{c}{L}\}$ | $o(\frac{\epsilon}{L\sqrt{d}})$ |
| ZO-SPIDER-coord (Ji et al.) | NC-S | ✓ | ✗ | $\mathcal{O}(d\sqrt{n}\epsilon^{-2})$ | $\frac{1}{\sqrt{n}L}$ | $\frac{1}{\sqrt{T}dL}$ |
| **PF-VRZO-coord** (Theorem 1) | NC-S | ✓ | ✓ | $\tilde{\mathcal{O}}(d\sqrt{n}\epsilon^{-2})$ | $\frac{1}{n^{1/4}\sqrt{(n^{1/2}+\sum_{s=0}^t\|v_s\|^2)}}$ | $\frac{1}{(t+1)\sqrt{nd}}$ |
| **PF-VRZO-rand** (Theorem 2) | NC-S | ✓ | ✓ | $\tilde{\mathcal{O}}(d\sqrt{n}\epsilon^{-2})$ | $\frac{1}{n^{1/4}\sqrt{d(n^{1/2}+\sum_{s=0}^t\|v_s\|^2)}}$ | $\frac{1}{(t+1)d\sqrt{n}}$ |

The key contributions of this work are as follows:

- **A Parameter-Free Zeroth-Order Framework:** We propose PF-VRZO, the first parameter-free zeroth-order optimization method for nonconvex finite-sum problems. It requires only minimal inputs—sample size $n$ and dimension $d$, without relying on problem-dependent parameters such as the smoothness constant or iteration count.

- **Variance Reduction with Adaptive Gradient Estimation:** PF-VRZO incorporates variance reduction into both coordinate-wise and random-direction zeroth-order estimators, with adaptive adjustment of smoothing parameters and step sizes, eliminating the need for manual tuning.

- **Theoretical and Empirical Validation:** We provide convergence guarantees showing that PF-VRZO achieves a function query complexity $\tilde{\mathcal{O}}(d\sqrt{n}\epsilon^{-2})$. Experiments on nonconvex phase retrieval and distributional robust optimization confirm its comparable performance compared to existing tuned methods.

## 2 PRELIMINARIES

**Remark 1.** *By "param-free," we mean that the method does not require any tunable hyperparameters—no manual adjustment is needed. The algorithm only depends on the dataset size $n$ and the model dimension $d$, both of which are inherent to the problem setup and readily available before running the optimization.*

**Notation** Throughout the paper, $\|\cdot\|$ denotes the Euclidean norm for vectors, $\tilde{\mathcal{O}}$ hide the logarithmic factors, and $\langle\cdot,\cdot\rangle$ denotes the inner product. We denote by $d$ the dimension of the problem, and by $n$ the number of functions in the optimization problem. We use $f_i(x)$ to denote the $i$-th sample function of $f(x)$.

**Definition 1** (Smoothness). *A function $f : \mathbb{R}^d \to \mathbb{R}$ is $L$-smooth if there exists $L > 0$ such that for all $x, y \in \mathbb{R}^d$:*

$$f(y) \leq f(x) + \langle \nabla f(x), y - x \rangle + \frac{L}{2}\|y - x\|^2$$

**Assumption 1** (Lipschitz Gradient). *Each function $f_i : \mathbb{R}^d \to \mathbb{R}$ is $L$-smooth such that*

$$\|\nabla f_i(\mathbf{x}) - \nabla f_i(\mathbf{y})\| \leq L\|\mathbf{x} - \mathbf{y}\|.$$

**Assumption 2** (Boundedness). *Let $f : \mathbb{R}^d \to \mathbb{R}$ be bounded from below by a finite constant $f^*$, i.e.,*

$$f(x_0) - f^* \leq \Delta.$$

*for the initial solution $x_0$.*

## 3 PROPOSED PARAMETER FREE VARIANCE REDUCED ZEROTH-ORDER METHODS

PF-VRZO(coord) method integrates variance reduction with zeroth-order gradient estimation in a parameter-free manner. This adaptive structure ensures stable updates and effective convergence, even in nonconvex settings.

To set the stage for our proposed PF-VRZO algorithm, we first review the fundamentals of zeroth-order optimization, followed by a summary of the main techniques introduced in this paper.

### 3.1 ZEROTH-ORDER GRADIENT ESTIMATORS

When the gradient of $f(x)$ is not directly obtainable, it is often estimated via coordinate-wise methods or Gaussian smoothing (Duchi et al., 2015; Gasnikov et al., 2023; Kornowski & Shamir, 2024; Lin et al., 2022). In what follows, we first describe the coordinate-wise estimator:

$$\bar{\nabla}_\mu f(x) := \sum_{\ell=1}^{d} \frac{1}{\mu} \left[ f\left(x + \mu \mathbf{e}_\ell\right) - f\left(x\right) \right] \mathbf{e}_\ell, \qquad \text{(Coord estimator)}$$

where $\mathbf{e}_l$ is a standard basis vector with $1$ at its $l^{\text{th}}$ coordinate, and $0$s elsewhere. The error of the coordinate-wise gradient estimator is upper bounded as follows, and it approaches zero as $\mu \to 0$ (Gao et al., 2018).

Besides the coordinate-wise estimator, the random-direction estimator is another widely used zeroth-order method, before introduce random-direction estimator, we first introduce smoothing function $f_\mu(x) := \mathbb{E}_{\{w \sim U_b\}}[f(x + \mu w)]$, where $U_b$ is a uniform distribution over the unit Euclidean ball, following Gao et al. (2018), its gradient can be expressed as $\nabla f_\mu(x) := \mathbb{E}_{\{\rho \sim U_{S_p}\}} \left[ \frac{n}{\mu} f(x + \mu \rho) \rho \right]$. Here $U_{S_p}$ is a uniform distribution over the unit Euclidean sphere, and $\rho \in \mathbb{R}^d$ is a random vector sampled from unit Euclidean sphere $U_{S_p}$. Now we can define zeroth-order random-direction estimator $\hat{\nabla} f(x)$ as follows, which is an unbiased estimator of $\nabla f_\mu(x)$:

$$\hat{\nabla}_\mu f(x) := \frac{d}{\mu} [f(x + \mu \rho) - f(x)] \rho. \qquad \text{(Random-direction estimator)}$$

Random-direction estimator is an unbiased estimate of the gradient of the smoothing function , i.e, $\mathbb{E}[\hat{\nabla}_\mu f(x)] = \nabla f_\mu(x)$.

Both of the aforementioned zeroth-order estimators rely on a fixed smoothing parameter $\mu$, whose improper tuning may lead to substantially degraded performance, ranging from slow convergence to divergence (Figure 1). To overcome this limitation, we develop a framework that integrates three key components: variance reduction, adaptive stepsize, and adaptive smoothing parameter. The latter two, in particular, set our method apart from existing approaches and enable new convergence guarantees.

### 3.2 VARIANCE REDUCTION TECHNIQUE

As a celebrated technique in stochastic optimization, variance reduction has been instrumental in the development of algorithms with significantly reduced theoretical complexity, SPIDER(Fang et al., 2018) is a variance reduction-typed method with optimal complexity guarantee, which uses large batch and small batch alternately to estimate stochastic gradients in a recursive way as follows:

$$v_t = \nabla f_B \left( \mathbf{x}^t \right) - \nabla f_B \left( \mathbf{x}^{t-1} \right) + v^{t-1}, \quad \text{(SPIDER)}$$

with clipped step size $\eta_t = \min\{c_1, \frac{c_2 \epsilon}{\|v_t\|}\}$ , where $c_1, c_2$ are some constants, and $\nabla f_B(x) = \frac{1}{|B|} \sum_{\xi \in B} \nabla f(x)$ with a small batch size $B$.

### 3.3 ADAPTIVE STEPSIZE

The step size $\gamma_t$ in PF-VRZO is chosen in a parameter-free and adaptive manner. Specifically, it is set as:

$$\gamma_t = \frac{1}{n^{1/4} c \sqrt{n^{1/2} + \sum_{s=0}^{t} \|v_s\|^2}},$$

We set $c = 1$ when using the coordinate-wise estimator and $c = \sqrt{d}$ when using the random-direction estimator. This design avoids reliance on unknown constants such as the Lipschitz constant or desired accuracy. By incorporating the accumulated gradient norms, the step size automatically decays, which helps balance exploration and convergence.

### 3.4 ADAPTIVE SMOOTHING PARAMETER

In PF-VRZO, the smoothing parameter $\mu_t$ plays a critical role in estimating gradients via zeroth-order information. Unlike traditional methods that fix $\mu$ based on prior knowledge of the target accuracy $\epsilon$ or total iterations $T$, PF-VRZO adaptively sets:

$$\mu_t = \frac{1}{(t+1)\sqrt{nd}}(Coord) \ , \ \mu_t = \frac{1}{(t+1)d\sqrt{n}}(Random)$$

which decreases over time. This schedule ensures that early iterations benefit from smoother approximations for stability, while later iterations use finer estimates for improved accuracy. The adaptive design of $\mu_t$ eliminates the need for manual tuning and allows the algorithm to adjust automatically throughout the optimization process.

### 3.5 PARAMETER-FREE VARIANCE REDUCED ZEROTH-ORDER METHOD(COORDWISE)

---

**Algorithm 1** PF-VRZO(coord)

---

Set $c = 1$ for coordwise estimator, $\mu_{-1} = \mu_0$.
**for** $t = 0$ **to** $T-1$ **do**
    Compute $\mu_t = \frac{1}{(t+1)\sqrt{nd}}$
    **if** $t \bmod n = 0$ **then**
        $v_t = \bar{\nabla}_{\mu_t} f(x_t)$ {Full zeroth-order gradient computation}
    **else**
        Uniformly sample $i_t \in \{1, \ldots, n\}$
        Compute $\bar{\nabla}_{\mu_t} f_{i_t}(x_t)$ with $\mu_t$ and $\bar{\nabla}_{\mu_{t-1}} f_{i_t}(x_{t-1})$ with $\mu_{t-1}$ .
        $v_t = \bar{\nabla}_{\mu_t} f_{i_t}(x_t) - \bar{\nabla}_{\mu_{t-1}} f_{i_t}(x_{t-1}) + v_{t-1}$
    **end if**
    $\gamma_t = \dfrac{1}{n^{1/4}c\sqrt{(n^{1/2} + \sum_{s=0}^{t}\|v_s\|^2)}}$
    $x_{t+1} = x_t - \gamma_t v_t$
**end for**

---

**Explanation of Algorithm 1:** For the constant $c$, we set $c = 1$ in this algorithm (which uses the coord estimator) and $c = \sqrt{d}$ in the algorithm with the rand estimator. At each iteration, the algorithm adaptively adjusts the smoothing parameter $\mu_t = 1/(t+1)\sqrt{nd}$, allowing finer gradient estimates as optimization progresses. Every $n$ iterations, a full zeroth-order gradient is computed as mentioned in Coord estimator. For the remaining steps, a variance-reduced estimator $v_t$ is constructed by combining the current and previous stochastic gradient estimates with $v_{t-1}$. The step size $\gamma_t$ is also adaptively computed based on the historical norm of the gradient estimates, eliminating the need for manual tuning.

---

To establish the convergence of our method, we divide the analysis into three parts.

$$\frac{1}{T}\mathbb{E}[\sum_{t=0}^{T-1}\|\nabla f(x_t)\|] \leq \frac{1}{T}[\underbrace{\sum_{t=0}^{T-1}\mathbb{E}[\|v_t\|]}_{\text{part I}} + \underbrace{\sum_{t=0}^{T-1}\mathbb{E}[\|v_t - \bar{\nabla}_{\mu_t} f(x_t)\|]}_{\text{part II}} + \underbrace{\sum_{t=0}^{T-1}\|\bar{\nabla}_{\mu_t} f(x_t) - \nabla f(x_t)\|]}_{\text{part III}}.$$

For each of these parts, we now present the corresponding lemmas (the detailed proofs can be found in Appendix C). Let $\delta_t := \frac{\sqrt{d}L\mu_t}{2}$ denote the error coefficient of the zeroth-order estimator. Then, we can derive the following results for Algorithm 1. First, we introduce a preliminary bound that will be repeatedly used in the subsequent analysis. The following lemma provides a bound that is frequently

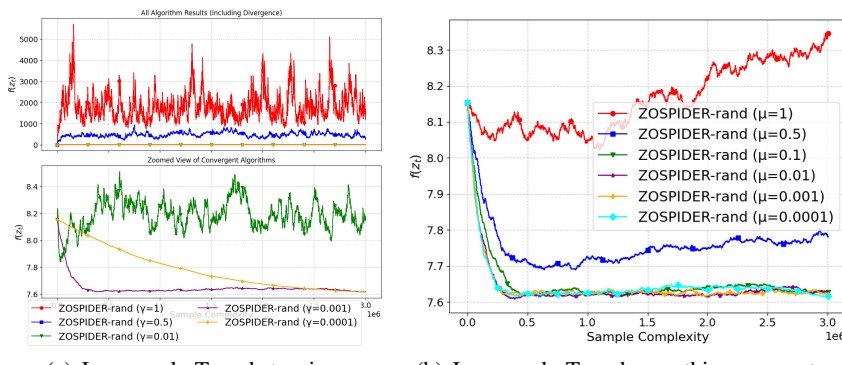

(a) Improperly Tuned stepsize        (b) Improperly Tuned smoothing parameter

Figure 1: This figure demonstrates the detrimental effects of improper parameter tuning on the optimization process through two subfigures. In (a), an improperly tuned stepsize leads to phenomena such as erratic fluctuations (e.g., the red curve in the upper subplot of (a)) and even non-convergence, while properly tuned stepsizes enable stable convergence (lower subplot of (a)). In (b), an improperly tuned smoothing parameter (e.g., $\mu = 1$ in the red curve) causes the optimization process to fail to converge, whereas appropriately tuned values (e.g., $\mu = 0.0001$) allow for effective convergence. Collectively, these results indicate that improperly tuned parameters can severely impair the optimization process, and in severe cases, even lead to non-convergence.

used in the proof. Although it may seem somewhat large, there is no need to worry because it will appear in logarithmic form in the proof.

**Lemma 1.** *Under assumptions 1 and 2, we have*

$$\sum_{t=0}^{T-1} \|v_t\|^2 \le \Phi(T) + 1.$$

*where $\Phi(T) := \frac{4TL^2 n^{1.5}}{c^2} + (32n^2 + 6) \sum_{t=0}^{T-1} \delta_t^2 + \frac{6L^2 T}{nc^2} + 6T \|\nabla f(x_0)\|^2 - 1$.*

Next, we provide an upper bound for each part separately.To facilitate the analysis, we transform the problem of the average gradient into two components: the gradient estimator $v_t$ (Part I) and the average of gradient estimation errors. The estimation error can be further decomposed into two parts: one is the error incurred by $v_t$ estimating the zeroth-order estimator $\bar{\nabla}_{\mu_t} f(x_t)$ (Part II), and the other is the error arising from replacing the true gradient $\nabla f(x_t)$ with the zeroth-order estimator (Part III). The following lemma aims to provide an upper bound for the SPIDER estimator $v_t$. Due to the complexity of this problem, we split the analysis into two lemmas.

**Lemma 2** (Part I(1)). *Under assumptions 1 and 2, we have*

$$\mathbb{E}\left[\sum_{t=0}^{T-1} \|v_t\|\right] \le n^{1/4}\sqrt{T}\left(2\Delta c + 2c\sum_{t=0}^{T-1} \gamma_t \delta_t^2 + 1 + \frac{L}{c}\log(\Phi(T)) + c \cdot \mathbb{E}\left[\sum_{t=0}^{T-1} \gamma_t \|v_t - \bar{\nabla}_{\mu_t} f(x_t)\|^2\right]\right).$$

The following lemma provides an upper bound for the last term in Part I(1).

**Lemma 3** (Part I(2)). *Under assumptions 1 and 2, we have*

$$\mathbb{E}\left[\sum_{t=0}^{T-1} \gamma_t \cdot \|v_t - \bar{\nabla}_{\mu_t} f(x_t)\|^2\right] \le \frac{2L^2}{c^3}\log(\Phi(T)) + \sum_{t=0}^{T-1} 16n\gamma_t \delta_t^2.$$

For the error incurred by the estimator $v_t$ in estimating the zeroth-order estimator $\bar{\nabla}_{\mu_t} f(x_t)$, we present the following lemma:

**Lemma 4** (Part II). *Under assumptions 1 and 2, we have*

$$\frac{1}{T}\mathbb{E}\left[\sum_{t=0}^{T-1} \|v_t - \bar{\nabla}_{\mu_t} f(x_t)\|\right] \le \frac{Ln^{1/4}}{c\sqrt{T}}\log(\Phi(T)) + \frac{1}{\sqrt{T}}\sqrt{8n\sum_{t=0}^{T-1} \delta_t^2}.$$

Based on the properties of the coordinate-wise zeroth-order estimator, we can directly give the upper bound for Part III as follows.

**Lemma 5** (Part III). *Under assumptions 1 and 2, we have* $\frac{1}{T}\sum_{t=0}^{T-1}\|\bar{\nabla}_{\mu_t}f(x_t) - \nabla_{\mu_t}f(x_t)\| \leq \frac{1}{T}\sum_{t=0}^{T-1}\delta_t$.

**Theorem 1** (Converge result of PF-VRZO(coord)). *Under assumptions 1, 2, we can derive the following result for Algorithm 1:*

$$\frac{1}{T}\mathbb{E}[\sum_{t=0}^{T-1}\|\nabla f(x_t)\|]$$

$$\leq \frac{n^{1/4}}{\sqrt{T}}\left(2\Delta \cdot c + 1 + (\frac{L}{c} + \frac{L^2}{c^2})\log(\Phi(T)) + \frac{L^2\pi^2}{24n^{\frac{1}{4}}} + \sqrt{\frac{\pi^2}{24}\frac{L}{n^{\frac{1}{4}}} + \frac{L^2\pi^2}{12}} + \frac{L}{2}\right)$$

*By setting $c = 1$, we can find stationary points of $f(x)$ with $T = \tilde{\mathcal{O}}(\sqrt{n}\epsilon^{-2})$.*

$$\frac{1}{T}\mathbb{E}[\sum_{t=0}^{T-1}\|\nabla f(x_t)\|] \leq \frac{1}{T}[\underbrace{\sum_{t=0}^{T-1}\mathbb{E}[\|v_t\|]}_{part\ I} + \underbrace{\sum_{t=0}^{T-1}\mathbb{E}[\|v_t - \bar{\nabla}_{\mu_t}f(x_t)\|]}_{part\ II} + \underbrace{\sum_{t=0}^{T-1}\|\bar{\nabla}_{\mu_t}f(x_t) - \nabla f(x_t)\|]}_{part\ III}$$

$$\leq \frac{n^{1/4}}{\sqrt{T}}\left(2\Delta \cdot c + 1 + (\frac{L}{cn^{\frac{3}{4}}} + \frac{L^2}{c^2})\log(\Phi(T))\right)$$

$$+ \frac{1}{T}(2c\sum_{t=0}^{T-1}\gamma_t\delta_t^2 + \sum_{t=0}^{T-1}\delta_t) + \frac{1}{\sqrt{T}}\left(n^{\frac{5}{4}}\sum_{t=0}^{T-1}c\gamma_t\delta_t^2 + n^{\frac{1}{2}}\sqrt{\sum_{t=0}^{T-1}2\delta_t^2}\right).$$

*Take $\delta_t = \frac{L}{2\sqrt{n}(t+1)}$ i.e.($\mu_t = \frac{1}{\sqrt{n}d}(t+1)$) then we can give an upper bound of $\sum_{t=0}^{T-1}\delta_t^2$ and $\sum_{t=0}^{T-1}\delta_t$ as follows:*

$$\sum_{t=0}^{T-1}\delta_t \leq \frac{LlnT}{2\sqrt{n}}, \quad \sum_{t=0}^{T-1}\delta_t^2 \leq \frac{L^2\pi^2}{24n}.$$

*With some calculations, we can obtain the final result.*

**Remark 2** (Discussion on the complexity). *Each coordwise estimator zeroth-order gradient estimation requires $\mathcal{O}(d)$ function evaluations. And since SPIDER consumes, on average, $\mathcal{O}(1+n/n)$ zeroth-order estimators per iteration, multiplying this by the total number of iterations $T = \tilde{\mathcal{O}}(\sqrt{n}\epsilon^{-2})$ yields a total function query complexity of $\#Function = \tilde{\mathcal{O}}\left(d\left(1 + \frac{n}{n}\right)T\right) = \tilde{\mathcal{O}}(d\sqrt{n}\epsilon^{-2})$.*

### 3.6 PROPOSED PARAMETER-FREE VARIANCE REDUCED ZEROTH-ORDER METHOD(RANDOM-DIRECTION ESTIMATOR)

In contrast to the coordinate-wise approach, which requires $\mathcal{O}(d)$ function evaluations per random estimator, the random method only incurs $\mathcal{O}(1)$ function evaluations per iteration. Nevertheless, it often requires $d$ times more iterations to achieve comparable accuracy. Therefore, the choice between the two methods can be made according to the practitioner's computational budget and application requirements. The analysis of the random-direction method follows essentially the same structure as that of the coordinate-wise method, although the final results differ slightly.

$$\frac{1}{T}\mathbb{E}[\sum_{t=0}^{T-1}\|\nabla f(x_t)\|] \leq \frac{1}{T}[\underbrace{\sum_{t=0}^{T-1}\mathbb{E}[\|v_t\|]}_{part\ I} + \underbrace{\sum_{t=0}^{T-1}\mathbb{E}[\|v_t - \nabla f_{\mu_t}(x)\|]}_{part\ II} + \underbrace{\sum_{t=0}^{T-1}\|\nabla f_{\mu_t}(x) - \nabla f(x_t),\|]}_{part\ III}$$

The proof of this part follows a similar argument as the coordinate estimator case and is therefore omitted. The complete proof can be found in Appendix D.

**Remark 3** (Proof Differences between the Random-direction and Coord Methods). *In the coordinate-wise method, we provide a bound $\mathcal{O}(\|x_t - x_{t-1}\|^2) + \mathcal{O}(\mu_t^2 + \mu_{t-1}^2)$ for the quantity $\|\bar{\nabla}_{\mu_t}f_{i_t}(x_t) - \bar{\nabla}_{\mu_{t-1}}f_{i_t}(x_{t-1})\|^2$. Although an estimation error exists, the smoothness of the coordinate estimator*

---

**Algorithm 2** PF-VRZO(Random-direction)

---

Set $c = \sqrt{d}$ for random-direction estimator and $\mu_{-1} = \mu_0$.
**for** $t = 0$ **to** $T-1$ **do**
    Compute smoothing parameter $\mu_t = \frac{1}{(t+1)d\sqrt{n}}$ , smoothing vector $\rho_t \sim U_B$.
    **if** $t \bmod n = 0$ **then**
      $v_t = \hat{\nabla}_{\mu_t} f(x_t)$ {Full zeroth-order gradient computation}
    **else**
      Sample $i_t \in \{1, \dots, n\}$ uniformly at random
      Compute $\hat{\nabla}_{\mu_t} f_{i_t}(x_t)$ with parameter $\mu_t$ and rand vector $\rho_t$, $\hat{\nabla}_{\mu_{t-1}} f_{i_t}(x_{t-1})$ with different parameter $\mu_{t-1}$ and the same rand vector $\rho_t$.
      $v_t = \hat{\nabla}_{\mu_t} f_{i_t}(x_t) - \hat{\nabla}_{\mu_{t-1}} f_{i_t}(x_{t-1}) + v_{t-1}$
    **end if**
    $\gamma_t = \dfrac{1}{n^{1/4}c\sqrt{(n^{1/2} + \sum_{s=0}^{t} \|v_s\|^2)}}$
    $x_{t+1} = x_t - \gamma_t v_t$
**end for**

---

**Explanation of Algorithm 2** Algorithm 2 shares an overall structure with Algorithm 1, with key differences as follows: 1.The zeroth-order estimator employs a random-direction estimator as mentioned in Random-direction estimator, where random numbers distributed on the unit sphere are generated by first sampling from a $d$-dimensional Gaussian distribution and then normalizing the sample. 2.We set $c = \sqrt{d}$ and use a smoothing parameter $\mu_t = \frac{1}{(t+1)d\sqrt{n}}$, introducing constant differences (involving $\sqrt{d}$) compared to the coordinate-wise variant ,where $c = 1$ and $\mu_t = \frac{1}{(t+1)\sqrt{dn}}$

---

*remains roughly of the same order as that of $f(x)$. In contrast, for the random-direction method, we obtain the estimate $\|\hat{\nabla}_{\mu_t} f(x_t) - \hat{\nabla}_{\mu_{t-1}} f(x_{t-1})\|^2 \leq \mathcal{O}(\mu_t^2 + \mu_{t-1}^2) + \mathcal{O}(d\|x_t - x_{t-1}\|^2)$, which suggests—albeit informally—that the smoothness of the random estimator is approximately $d$ times larger than that of $f(x)$. This distinction is reflected in the conclusions of various parts of the analysis, and, in particular, it necessitates choosing $c = \sqrt{d}, \mu_t = \frac{1}{(t+1)d\sqrt{n}}$ in the proof of the theorem (whereas $c = 1, \mu_t = \frac{1}{(t+1)\sqrt{dn}}$ suffices in the coordinate-wise case). As a result, the number of iterations required by the random-direction method is $d$ times larger than that of the coord method.*

**Theorem 2** (Converge result of PF-VRZO(random-direction). *Under assumptions 1, 2, we can derive the following result for Algorithm 2:*

$$\frac{1}{T}\mathbb{E}[\sum_{t=0}^{T-1} \|\nabla f(x_t)\|]$$

$$\leq \frac{n^{1/4}}{\sqrt{T}}\left(\Delta \cdot c + 1 + (\frac{L\sqrt{d}}{cn^{\frac{3}{4}}} + \frac{L^2 d}{c^2})\log(\phi(T)) + \frac{L^2\pi^2}{24n^{\frac{1}{4}}} + \frac{L}{n^{\frac{1}{4}}}\sqrt{\frac{\pi^2}{24}} + \frac{L^2\pi^2}{12} + \frac{L}{2}\right)$$

*By setting $c = \sqrt{d}, \mu_t = \frac{1}{d\sqrt{n}}(t+1)$, we can find stationary points of $f(x)$ with $T = \tilde{\mathcal{O}}(d\sqrt{n}\epsilon^{-2})$.*

**Remark 4** (Discussion on the complexity). *Each Random-direction zeroth-order gradient estimation requires $\mathcal{O}(1)$ function evaluations.And since SPIDER consumes, on average, $\mathcal{O}(1 + n/n)$ zeroth-order estimators per iteration, multiplying this by the total number of iterations $T = \tilde{\mathcal{O}}(d\sqrt{n}\epsilon^{-2})$ yields a total function query complexity of $\#Function = \tilde{\mathcal{O}}\left((1 + \frac{n}{n})T\right) = \tilde{\mathcal{O}}(d\sqrt{n}\epsilon^{-2})$.*

## 4 EXPERIMENTS

We conduct two experiments to evaluate the effectiveness of our method: the first focuses on Phase Retrieval, as shown in Figures 2(a) and 2(b), while the second examines Distributional Robust Optimization (DRO), presented in Figures **??** and **??**. To validate the performance of our algorithm, we compare it with ZO-SPIDER(Ji et al., 2019) and ZO-SGD (Ghadimi & Lan, 2013), both of which rely on manually tuned hyperparameters to ensure convergence, in contrast to our parameter-free

approach. We measure computational cost using both sample complexity and time. Here, sample complexity refers to the total number of function value evaluations. Due to space limitations, we defer the detailed descriptions of the hyperparameter settings to Appendix E. All experiments are conducted on a single NVIDIA RTX 3090 GPU.

## 4.1 APPLICATION TO NONCONVEX PHASE RETRIEVAL

Phase retrieval is a well-known nonconvex problem in machine learning and signal processing(Miao et al., 1999). Let $x \in \mathbb{R}^d$ represent the true underlying object, and assume we collect $m$ intensity measurements, given by $y_r = |\mathbf{a}_r^\top x|^2$ for $r = 1, 2, \ldots, m$, where $\mathbf{a}_r \in \mathbb{R}^d$. The challenge in phase retrieval lies in recovering the signal by solving the associated nonconvex optimization problem:

$$\min_{z \in \mathbb{R}^d} f(z) := \frac{1}{2m} \sum_{r=1}^{m} \left( y_r - |\mathbf{a}_r^\top z|^2 \right)^2. \tag{2}$$

We assess the effectiveness of our algorithms on the nonconvex phase retrieval task defined in (2). As illustrated in Figures 2(a) and 2(b), the proposed PF-VRZO algorithm demonstrates robust performance, notably without requiring manual tuning of algorithmic parameters.

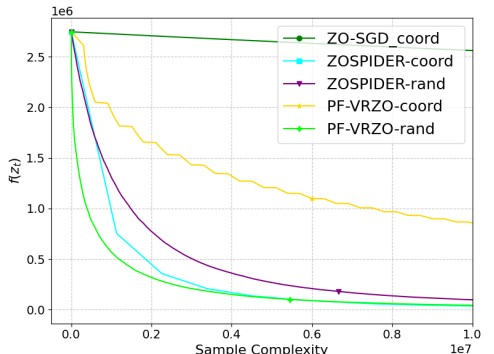
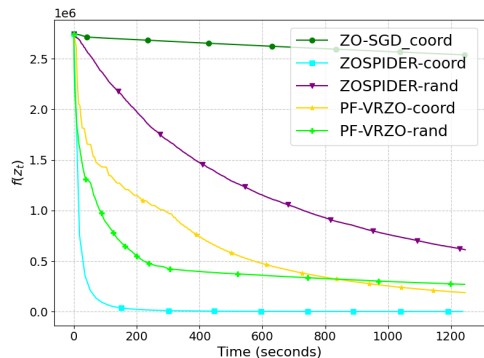

(a) Compare different algorithms on Phase Retrieval(Complexity)

(b) Compare different algorithms on Phase Retrieval(Time)

Figure 2: This figure compares the performance of different algorithms on Phase Retrieval through two subfigures. In (a), we evaluate the sample complexity of algorithms including PF-VRZO-coord, PF-VRZO-rand, ZO-SGD-coord, ZOSPIDER-coord, and ZOSPIDER-rand. In (b), we assess their time efficiency. Notably, the proposed PF-VRZO method, even without parameter tuning, demonstrates competitive performance when compared to other algorithms that undergo multiple parameter adjustments, indicating its robustness and effectiveness in Phase Retrieval tasks.

## 4.2 APPLICATION TO DISTRIBUTIONAL ROBUST OPTIMIZATION

Distributional Robust Optimization (DRO) is a widely used framework for training robust models, Under mild conditions, it aims to solve the following problem:

$$\min_{x \in \mathcal{X}, \eta \in \mathbb{R}} L(x, \eta) := \lambda \mathbb{E} \xi \sim P \psi^* \left( \frac{\ell \xi(x) - \eta}{\lambda} \right) + \eta \tag{3}$$

We consider the nonconvex DRO problem (3) on three real-world datasets. The Life Expectancy dataset contains 2,413 samples with 20 associated features. The Communities and Crime dataset consists of 1,994 samples and 122 predictive features. The Arcene dataset includes 200 samples with 10,000 high-dimensional features, making it a challenging benchmark for robust optimization.After standard preprocessing steps, including missing value imputation and variable standardization, we retain 70% samples for training, where each input $x_i \in \mathbb{R}^{34}$ and corresponding target $y_i \in \mathbb{R}$. We set the regularization parameter to $\lambda = 0.01$, and adopt the $\chi^2$-divergence, with the convex conjugate

given by $\psi^*(t) = \frac{1}{4}(t+2)^2 - 1$. The regularized loss function is defined as:

$$\ell_\xi(w) = \frac{1}{2}(y_\xi - x_\xi^\top w)^2 + 0.1 \sum_{j=1}^{34} \ln\left(1 + |w^{(j)}|\right).$$

We initialize $w_0 \in \mathbb{R}^{34}$ from a Gaussian distribution and set the initial step size $\eta_0 = 0.1$.

Based on the experimental results shown in Figures 3, we observe that the proposed PF-VRZO method exhibits a brief oscillation in the objective value at the beginning, likely due to insufficient accumulated gradient information. However, the method quickly resumes descent and ultimately achieves strong performance without the need for any parameter tuning.

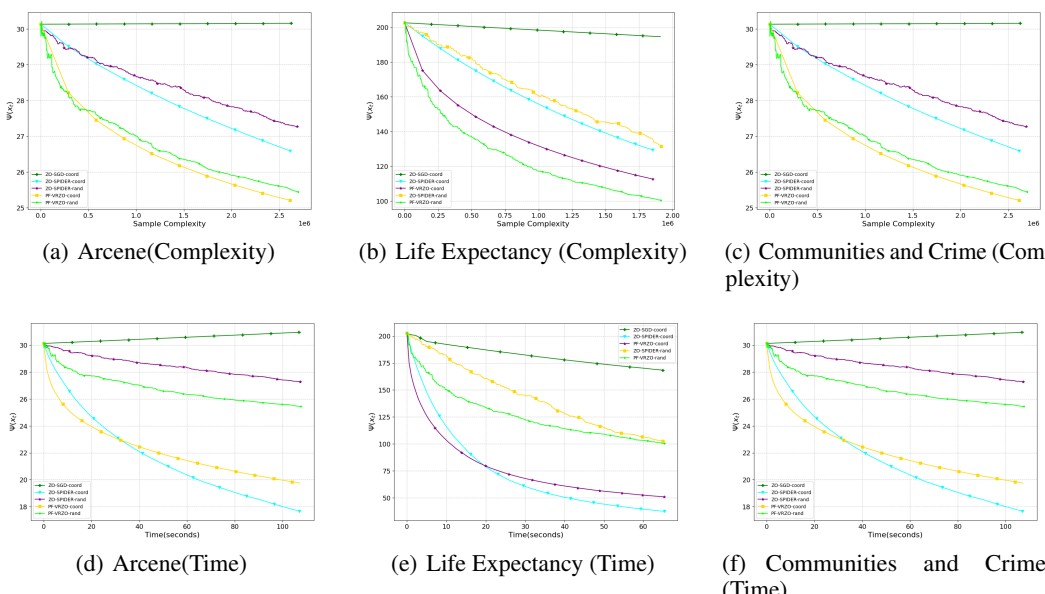

(a) Arcene(Complexity)

(b) Life Expectancy (Complexity)

(c) Communities and Crime (Complexity)

(d) Arcene(Time)

(e) Life Expectancy (Time)

(f) Communities and Crime (Time)

Figure 3: This figure evaluates the performance of different algorithms on Distributionally Robust Optimization (DRO) tasks across three datasets (Arcene, Life Expectancy, Communities and Crime), with results split into two metrics: Sample Complexity (subfigures (a)-(c)): Measures the number of samples required for algorithms to converge. Time Efficiency (subfigures (d)-(f)): Measures the runtime (in seconds) for algorithms to converge. Across all datasets and metrics, the proposed methods (e.g., PF-VRZO variants) demonstrate competitive or superior performance—consistently achieving faster convergence . This further validates the effectiveness of the parameter-free design of PF-VRZO in practical DRO scenarios.

## 5 CONCLUSION

In this paper, we propose a parameter-free variance-reduced zeroth-order method (PF-VRZO) for nonconvex optimization. Our method is based on the SPIDER framework and employs a coordinate-wise or random-direction zeroth-order gradient estimator. We establish the convergence of our method, demonstrating that it achieves a sample complexity of $\tilde{\mathcal{O}}(d\sqrt{n}\epsilon^{-2})$ for finding stationary points of nonconvex functions. Additionally, we conduct experiments on nonconvex phase retrieval and distributionally robust optimization to validate the effectiveness of our method. An interesting future direction is to investigate whether the logarithmic, $L$-dependent, and $\Delta$-dependent terms in the complexity bounds are optimal. (Carmon & Hinder, 2024) shows that under the convex-but-nonsmooth (C-NS) setting, any adaptive algorithm necessarily suffers from worse complexity. However, it remains unclear whether a similar conclusion holds under the nonconvex-smooth (NC-S) setting.

# 6 ETHICS STATEMENT

Our study focuses on developing a novel optimization algorithm and does not involve human subjects, animal experimentation, or the use of sensitive personal data. All experiments are conducted on publicly available datasets that are commonly used within the academic community. We adhere to the ICLR Code of Ethics, and our work introduces no new privacy or ethical risks beyond those inherent in standard academic research on optimization methods.

# 7 REPRODUCIBILITY STATEMENT

We have made every effort to ensure the reproducibility of our results. The paper provides detailed specifications for our proposed algorithm, PF-VRZO, including its variants and their theoretical foundations. We have meticulously described our experimental setup, including the specific nonconvex problems we studied, the parameters used for all compared algorithms (e.g., learning rates and batch sizes for ZO-SGD, PF-VRZO, ZO-SPIDER), and the hardware used.

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

CONTENTS

## A  RELATED WORK

### A.1  ZEROTH-ORDER OPTIMIZATION

The ZO-SGD method was first introduced by (Ghadimi & Lan, 2013), serving as a foundational approach in zeroth-order stochastic optimization. To enhance its efficiency, several follow-up works (Liu et al., 2018a) proposed accelerated variants, collectively referred to as ZO-SVRG, which incorporate the SVRG framework (Johnson & Zhang, 2013). In addition, to further reduce the function query complexity, ZO-SPIDER-Coord (Ji et al., 2019) were developed based on the stochastic path-integrated differential estimator.

### A.2  PARAMETER-FREE OPTIMIZATION

Recent advances in the nonconvex and smooth setting have drawn inspiration from AdaGrad, as introduced in the concurrent seminal works (Duchi et al., 2011; McMahan & Streeter, 2010). Building on this foundation, (Kavis et al., 2022) were the first to develop a parameter-free algorithm that remains compatible with variance reduction techniques. This was later improved by (Jiang et al., 2024), who proposed ADA-STORM, reducing the overall complexity by a logarithmic factor. Moreover, a series of follow-up studies (Ivgi et al., 2023; Kreisler et al., 2024; Orabona & Tommasi, 2017; Chen et al., 2022; Defazio & Mishchenko, 2023) have explored parameter-free methods in various problem settings, and other works have investigated the fundamental lower bounds of such algorithms (Khaled & Jin, 2024; Attia & Koren, 2024; Carmon & Hinder, 2024).

## B  USEFUL FACTS

**Lemma B.1** (Jensen's inequality). *For convex function $f(x)$ we have*

$$f(\mathbb{E}[x]) \leq \mathbb{E}[f(x)],$$

*two extended versions of Jensen's inequality are*

$$\|\mathbb{E}[x]\| \leq \mathbb{E}[\|x\|], \text{ for } x \in \mathbb{R}^d$$

$$\left\|\sum_{i=1}^{k} a_i\right\|^2 \leq k \sum_{i=1}^{k} \|a_i\|^2, \text{ for } a_i \in \mathbb{R}^d.$$

**Lemma B.2** (Young's inequality). *For any vectors $a, b, \in \mathbb{R}^d$, and $\zeta \geq 0$, the following inequality holds:*

$$\|a\|^2 \leq (1+\zeta)\|a-b\|^2 + \left(1+\zeta^{-1}\right)\|b\|^2,$$

*an extended version of Young's inequality is*

$$\langle a, b \rangle \leq \frac{\|a\|^2}{2\zeta} + \frac{\zeta\|b\|^2}{2}.$$

**Lemma B.3** (variance decomposition). *For random vector $x \in \mathbb{R}^d$ and any $y \in \mathbb{R}^d$, the variance of $x$ can be decomposed as*

$$\mathbb{E}\left[\|x - \mathbb{E}[x]\|^2\right] = \mathbb{E}\left[\|x - y\|^2\right] - \mathbb{E}\left[\|\mathbb{E}[x] - y\|^2\right],$$

*which implies*

$$\mathbb{E}\left[\|x - \mathbb{E}[x]\|^2\right] \leq \mathbb{E}\left[\|x\|^2\right].$$

**Lemma B.4.** *For random variable $X, Y$, if $X, Y$ are independent, and $\mathbb{E}[X]$ or $\mathbb{E}[Y] = 0$, we have*

$$\mathbb{E}[\|X - Y\|^2] = \mathbb{E}[\|X\|^2] + \mathbb{E}[\|Y\|^2].$$

*Proof.*

$$\mathbb{E}[\|X - Y\|^2] = \mathbb{E}[\|X\|^2 + \|Y\|^2 + 2\mathbb{E}\langle X, Y \rangle] = \mathbb{E}[\|X\|^2] + \mathbb{E}[\|Y\|^2].$$

$\square$

**Lemma B.5.** *For i.i.d. $x_1, x_2, x_3 \cdots x_n$, if $\mathbb{E}[x_i] = x, \mathbb{E}[\|x_i - x\|^2] \leq \sigma^2$, we have*

$$\mathbb{E}\left[\left\|\frac{1}{b}\sum_{i=1}^{b} x_i - x\right\|^2\right] \leq \frac{\mathbb{E}[\|x_i\|^2]}{b}.$$

*Proof.*

$$\mathbb{E}\left[\left\|\frac{1}{b}\sum_{i=1}^{b} x_i - x\right\|^2\right]$$

$$= \frac{1}{b^2}\mathbb{E}\left[\left\|\sum_{i=1}^{b}(x_i - x)\right\|^2\right]$$

$$= \frac{1}{b^2}\sum_{i=1}^{b}\mathbb{E}[\|x_i - x\|^2]$$

$$= \frac{1}{b}\mathbb{E}[\|x_i - x\|^2] \leq \frac{\mathbb{E}[\|x_i\|^2]}{b},$$

where the second inequality holds because $\|a + b\|^2 = \|a\|^2 + \|b\|^2 + 2\langle a, b \rangle$, and $\mathbb{E}[\langle x_i - x, x_j - x \rangle] = 0 (j \neq i)$ for iid random variable $x_i$. $\square$

**Lemma B.6** (Sum of Square Roots Inequality)**.** *Let $\alpha_1, \ldots, \alpha_T$ be a sequence of non-negative real numbers ($\alpha_t \geq 0$ for all t). Then:*

$$\sqrt{\sum_{t=1}^{T} \alpha_t} \leq \sum_{t=1}^{T} \frac{\alpha_t}{\sqrt{\sum_{s=1}^{t} \alpha_s}}.$$

**Lemma B.7** (Logarithmic Sum Bound)**.** *For any sequence of non-negative real numbers $a_1, \ldots, a_T$ with $a_1 \geq 1$, we have:*

$$\sum_{\ell=1}^{T} \frac{a_\ell}{1 + \sum_{i=1}^{\ell} a_i} \leq \log\left(\sum_{i=1}^{T} a_i + 1\right)$$

**Lemma B.8** (Sum of $\frac{1}{i}$ and $\frac{1}{i^2}$)**.**

$$\sum_{i=1}^{T-1} \frac{1}{i} \leq \log(T).$$

$$\sum_{i=1}^{\infty} \frac{1}{i^2} \leq \frac{\pi^2}{6}.$$

## C  PARAMETER FREE VARIANCE REDUCED ZEROTH-ORDER METHOD(COORD)

---

**Algorithm 3** PF-VRZO(coord)

---

Set $c = 1$ for coordwise estimator, $\mu_{-1} = \mu_0$.
**for** $t = 0$ **to** $T-1$ **do**
    Compute $\mu_t = \frac{1}{(t+1)\sqrt{nd}}$
    **if** $t \bmod n = 0$ **then**
      $v_t = \bar{\nabla}_{\mu_t} f(x_t)$ {Full zeroth-order gradient computation}
    **else**
      Uniformly sample $i_t \in \{1, \ldots, n\}$
      Compute $\bar{\nabla}_{\mu_t} f_{i_t}(x_t)$ with $\mu_t$ and $\bar{\nabla}_{\mu_{t-1}} f_{i_t}(x_{t-1})$ with $\mu_{t-1}$.
      $v_t = \bar{\nabla}_{\mu_t} f_{i_t}(x_t) - \bar{\nabla}_{\mu_{t-1}} f_{i_t}(x_{t-1}) + v_{t-1}$
    **end if**
    $\gamma_t = \dfrac{1}{n^{1/4} c \sqrt{(n^{1/2} + \sum_{s=0}^{t} \|v_s\|^2)}}$
    $x_{t+1} = x_t - \gamma_t v_t$
**end for**

---

Table 2: Meaning of Symbols

| Symbol | Meaning |
|---|---|
| $\gamma_t$ | stepsize $\frac{1}{\left(n^{1/4} c \sqrt{n^{1/2} + \sum_{s=0}^{t} \|v_s\|^2}\right)}$. |
| $\mu_t$ | Smoothing parameter at iteration $t$. |
| $v_t$ | Spider estimator. |
| $\bar{\nabla}_\mu f(x_t)$ | zeroth-order estimator(coord) . |
| $\delta_t$ | $\sqrt{d} L \mu_t / 2$, the estimation error with respect to $\bar{\nabla} f_\mu$. |

To establish the convergence of our method, we divide the analysis into three parts.

$$\frac{1}{T} \mathbb{E}[\sum_{t=0}^{T-1} \|\nabla f(x_t)\|] \leq \frac{1}{T}[\underbrace{\sum_{t=0}^{T-1} \mathbb{E}[\|v_t\|]}_{\text{part I}} + \underbrace{\sum_{t=0}^{T-1} \mathbb{E}[\|v_t - \bar{\nabla}_{\mu_t} f(x_t)\|]}_{\text{part II}} + \underbrace{\sum_{t=0}^{T-1} \|\bar{\nabla}_{\mu_t} f(x_t) - \nabla f(x_t)\|]}_{\text{part III}}.$$

**Lemma C.1** ((Gao et al., 2018)). *For L-smooth function $f(x)$, its gradient $\nabla f(x)$ and its coord zeroth-order estimator $\bar{\nabla}_\mu f(x)$, we have*

$$\left\| \bar{\nabla}_\mu f(x) - \nabla f(x) \right\|^2 \leq \delta_t^2.$$

*where $\delta_t := \sqrt{d} L \mu_t / 2$, and $\mu_t$ is the smoothing parameter at iteration $t$.*

**Lemma C.2.** *Under assumptions 1 and 2, we can derive the following result for Algorithm 1:*

$$\sum_{t=0}^{T-1} \|v_t\|^2 \leq \Phi(T) + 1.$$

*where $\frac{4TL^2 n^{1.5}}{c^2} + (32n^2 + 6) \sum_{t=0}^{T-1} \delta_t^2 + \frac{6L^2 T}{nc^2} + 6T \|\nabla f(x_0)\|^2 - 1$. Here the notation $\Phi(T)$ is introduced only for brevity, and will be repeatedly used in the subsequent analysis.*

*Proof.*

$$\|v_t\|^2 = \left\| \sum_{s=t-t \bmod n+1}^{t} \left( \bar{\nabla}_{\mu_s} f_{i_s}(x_s) - \bar{\nabla}_{\mu_{s-1}} f_{i_s}(x_{s-1}) \right) + \bar{\nabla}_{\mu_{t-t \bmod n}} f(x_{t-t \bmod n}) \right\|^2$$

$$\leq 2 \cdot \left\| \sum_{s=t-t \bmod n+1}^{t} \bar{\nabla}_{\mu_s} f_{i_s}(x_s) - \bar{\nabla}_{\mu_{s-1}} f_{i_s}(x_{s-1}) \right\|^2 + 2 \cdot \left\| \bar{\nabla}_{\mu_{t-t \bmod n}} f(x_{t-t \bmod n}) \right\|^2$$

$$\leq 2n \cdot \sum_{s=t-t \bmod n+1}^{t} \left\| \bar{\nabla}_{\mu_s} f_{i_s}(x_s) - \bar{\nabla}_{\mu_{s-1}} f_{i_s}(x_{s-1}) \right\|^2 + 2 \cdot \left\| \bar{\nabla}_{\mu_{t-t \bmod n}} f(x_{t-t \bmod n}) \right\|^2$$

$$\overset{lem C.1}{\leq} 2n \cdot \sum_{s=t-t \bmod n+1}^{t} 2L^2 (x_s - x_{s-1})^2 + 8(\delta_s^2 + \delta_{s-1}^2) + 2 \cdot \left\| \bar{\nabla}_{\mu_{t-t \bmod n}} f(x_{t-t \bmod n}) \right\|^2$$

$$\leq \frac{4L^2 n^{1.5}}{c^2} + 16n \sum_{s=t-t \bmod n+1}^{t} (\delta_s^2 + \delta_{s-1}^2) + 2 \cdot \left\| \bar{\nabla}_{\mu_{t-t \bmod n}} f(x_{t-t \bmod n}) \right\|^2,$$

in last inequality we use $\|x_s - x_{s-1}\| = \frac{1}{n^{1/4}c} * \underbrace{\left\| \frac{v_t}{\sqrt{\left( n^{1/2} + \sum_{s=0}^{t} \|v_s\|^2 \right)}} \right\|}_{\leq 1} \leq \frac{1}{n^{1/4}c}$. then we

bound $\bar{\nabla}_{\mu_t} f(x)$ below:

$$\begin{aligned}
\left\| \bar{\nabla}_{\mu_t} f(x_t) \right\| &\leq \left\| \bar{\nabla}_{\mu_t} f(x_t) - \nabla f(x_t) \right\| + \left\| \nabla f(x_t) \right\| \\
&\leq \delta_t + \left\| \nabla f(x_t) - \nabla f(x_0) \right\| + \left\| \nabla f(x_0) \right\| \\
&\leq L \left\| x_t - x_0 \right\| + \delta_t + \left\| \nabla f(x_0) \right\| \\
&\leq L \sum_{i=1}^{t} \left\| x_i - x_{i-1} \right\| + \delta_t + \left\| \nabla f(x_0) \right\| \\
&\leq \left( \frac{L}{c\sqrt{n}} \right) + \delta_t + \left\| \nabla f(x_0) \right\|.
\end{aligned}$$

Combining the above results we obtain(Without loss of generality, we set $\delta_{-1} = \delta_0$):

$$\begin{aligned}
\sum_{t=0}^{T-1} \|v_t\|^2 &\leq \sum_{t=0}^{T-1} \left( \frac{4L^2 n^{1.5}}{c^2} + 16n \sum_{s=t-t \bmod n+1}^{t} (\delta_s^2 + \delta_{s-1}^2) + 2 \cdot \left\| \bar{\nabla}_{\mu_{t-t \bmod n}} f(x_{t-t \bmod n}) \right\|^2 \right) \\
&\leq \frac{4TL^2 n^{1.5}}{c^2} + 16n \sum_{t=0}^{T-1} \sum_{s=t-t \bmod n+1}^{t} (\delta_s^2 + \delta_{s-1}^2) + 2 \sum_{t=0}^{T-1} \left\| \bar{\nabla}_{\mu_t} f(x_t) \right\|^2 \\
&\leq \frac{4TL^2 n^{1.5}}{c^2} + 32n^2 \sum_{t=0}^{T-1} \delta_t^2 + 2 \sum_{t=0}^{T-1} \left( \left( \frac{L}{c\sqrt{n}} \right) + \delta_t + \left\| \nabla f(x_0) \right\| \right)^2 \\
&\leq \frac{4TL^2 n^{1.5}}{c^2} + (32n^2 + 6) \sum_{t=0}^{T-1} \delta_t^2 + \frac{6L^2 T}{nc^2} + 6T \left\| \nabla f(x_0) \right\|^2.
\end{aligned}$$

Since this equation will be used repeatedly, we define $\Phi(T) := \frac{4TL^2 n^{1.5}}{c^2} + (32n^2 + 6) \sum_{t=0}^{T-1} \delta_t^2 + \frac{6L^2 T}{nc^2} + 6T \left\| \nabla f(x_0) \right\|^2 - 1$ to simplify the resulting expressions. $\qquad \square$

## C.1 PART I

**Lemma C.3** (part I(1)). *Under assumptions 1 and 2, we can derive the following result for Algorithm 1:*

$$\mathbb{E}\left[ \sum_{t=0}^{T-1} \|v_t\| \right] \leq n^{1/4} \sqrt{T} \left( 2\Delta c + 2c \sum_{t=0}^{T-1} \gamma_t \delta_t^2 + 1 + \frac{L}{c} \log(\Phi(T)) + c \cdot \mathbb{E}\left[ \sum_{t=0}^{T-1} \gamma_t \|\bar{\nabla}_{\mu_t} f(x_t) - v_t\|^2 \right] \right).$$

*Proof.*

$$\mathbb{E}\left[f(x_{t+1}) \mid \mathcal{F}_t\right] \leq \mathbb{E}\left[f(x_t) + \nabla f(x_t)^T(x_{t+1} - x_t) + \frac{L}{2}\|x_t - x_{t+1}\|^2 \mid \mathcal{F}_t\right]$$

$$= \mathbb{E}\left[f(x_t) - \gamma_t v_t^T \nabla f(x_t) + \frac{L}{2}\gamma_t^2\|v_t\|^2 \mid \mathcal{F}_t\right]$$

$$\leq \mathbb{E}\left[f(x_t) + \frac{\gamma_t}{2}\|v_t - \nabla f(x_t)\|^2 - \frac{\gamma_t}{2}(1 - L\gamma_t)\|v_t\|^2 \mid \mathcal{F}_t\right]$$

$$\overset{lem C.1}{\leq} \mathbb{E}\left[f(x_t) + \gamma_t \delta_t^2 + \gamma_t\|v_t - \bar{\nabla}_{\mu_t} f(x_t)\|^2 - \frac{\gamma_t}{2}(1 - L\gamma_t)\|v_t\|^2 \mid \mathcal{F}_t\right],$$

thus we obtain:

$$\mathbb{E}\left[\gamma_t \cdot \|v_t\|^2\right] \leq 2\mathbb{E}\left[f(x_t) - f(x_{t+1})\right] + 2\gamma_t \delta_t^2 + \mathbb{E}\left[L\gamma_t^2 \cdot \|v_t\|^2\right] + 2 \cdot \mathbb{E}\left[\gamma_t \cdot \|\bar{\nabla}_{\mu_t} f(x_t) - v_t\|^2\right].$$

By summing from $t = 0$ to $T - 1$ we get:

$$\sum_{t=0}^{T-1}\mathbb{E}\left[\gamma_t \cdot \|v_t\|^2\right] \leq 2\Delta + 2\sum_{t=0}^{T-1}\gamma_t \delta_t^2 + \mathbb{E}\left[\sum_{t=0}^{T-1}L\gamma_t^2 \cdot \|v_t\|^2\right] + \mathbb{E}\left[\sum_{t=0}^{T-1}2\gamma_t \cdot \|\bar{\nabla}_{\mu_t} f(x_t) - v_t\|^2\right].$$

Recall that $\gamma_t = n^{-1/4}c^{-1}\left(n^{1/2} + \sum_{s=0}^{t}\|v_s\|^2\right)^{-1/2}$ and Lemma B.7 we obtain:

$$\mathbb{E}\left[\sum_{t=0}^{T-1}\gamma_t \cdot \|v_t\|^2\right] \leq 2\Delta + 2\sum_{t=0}^{T-1}\gamma_t \delta_t^2 + \frac{L}{c^2\sqrt{n}} \cdot \mathbb{E}\left[\sum_{t=0}^{T-1}\frac{\|v_t\|^2}{\sqrt{n} + \sum_{s=0}^{t}\|v_s\|^2}\right]$$

$$+ 2\mathbb{E}\left[\sum_{t=0}^{T-1}\gamma_t \cdot \|\bar{\nabla}_{\mu_t} f(x_t) - v_t\|^2\right]$$

$$\leq 2\Delta + 2\sum_{t=0}^{T-1}\gamma_t \delta_t^2 + \frac{L}{c^2\sqrt{n}}\log(\Phi(T)) + 2\mathbb{E}\left[\sum_{t=0}^{T-1}\gamma_t \cdot \|\bar{\nabla}_{\mu_t} f(x_t) - v_t\|^2\right].$$

Lower bounding the right-hand side:

$$\mathbb{E}\left[\sum_{t=0}^{T-1}\gamma_t \cdot \|v_t\|^2\right] \geq \mathbb{E}\left[\frac{\sum_{t=0}^{T-1}\|v_t\|^2}{n^{1/4}c\sqrt{n^{1/2} + \sum_{t=0}^{T-1}\|v_t\|^2}}\right]$$

$$\geq \frac{1}{c} \cdot \mathbb{E}\left[\frac{\sum_{t=0}^{T-1}\|v_t\|^2/\sqrt{n}}{\sqrt{1 + \sum_{t=0}^{T-1}\|v_t\|^2/\sqrt{n}}}\right]$$

$$\geq \frac{1}{c} \cdot \left(\mathbb{E}\left[\sqrt{\sum_{t=0}^{T-1}\|v_t\|^2/\sqrt{n}}\right] - 1\right)$$

$$\geq \frac{1}{cn^{1/4}\sqrt{T}}\mathbb{E}\left[\sum_{t=0}^{T-1}\|v_t\|\right] - \frac{1}{c}.$$

Combining all results:

$$\mathbb{E}\left[\sum_{t=0}^{T-1}\|v_t\|\right] \leq n^{1/4}\sqrt{T}\left(2\Delta c + 2c\sum_{t=0}^{T-1}\gamma_t \delta_t^2 + 1 + \frac{L}{c\sqrt{n}}\log(\Phi(T)) + c \cdot \mathbb{E}\left[\sum_{t=0}^{T-1}\gamma_t\|\bar{\nabla}_{\mu_t} f(x_t) - v_t\|^2\right]\right).$$

$\square$

**Lemma C.4** (part I(2)). *Under assumptions 1 and 2, we can derive the following result for Algorithm 1:*

$$\mathbb{E}\left[\sum_{t=0}^{T-1}\gamma_t\cdot\|v_t-\bar{\nabla}_{\mu_t}f(x_t)\|^2\right]\leq\frac{2L^2}{c^3}\log(\Phi(T))+\sum_{t=0}^{T-1}16n\gamma_t\delta_t^2.$$

*Proof.* Let $\mathcal{F}_t$ be the sigma-algebra generated by $\{i_0,\ldots,i_t\}$ and $x_0$. From the definition of $\gamma_t$, it follows that $\gamma_t\leq\gamma_{t-1}$; this condition is imposed to resolve measurability concerns. Consequently,

$$\mathbb{E}\left[\gamma_t\|v_t-\bar{\nabla}_{\mu_t}f(x_t)\|^2\mid\mathcal{F}_{t-1}\right]\leq\mathbb{E}\left[\gamma_{t-1}\cdot\|v_t-\bar{\nabla}_{\mu_t}f(x_t)\|^2\mid\mathcal{F}_{t-1}\right].$$

Hence, our analysis can be reduced to studying $\mathbb{E}\left[\gamma_{t-1}\|v_t-\bar{\nabla}_{\mu_t}f(x_t)\|^2\mid\mathcal{F}_{t-1}\right]$.

$$\mathbb{E}\left[\gamma_{t-1}\|v_t-\bar{\nabla}_{\mu_t}f(x_t)\|^2\mid\mathcal{F}_{t-1}\right]$$
$$=\gamma_{t-1}\mathbb{E}\left[\|\bar{\nabla}_{\mu_t}f_{i_t}(x_t)-\bar{\nabla}_{\mu_{t-1}}f_{i_t}(x_{t-1})-\bar{\nabla}_{\mu_t}f(x_t)+\bar{\nabla}_{\mu_{t-1}}f(x_{t-1})+(v_{t-1}-\bar{\nabla}_{\mu_{t-1}}f(x_{t-1}))\|^2\mid\mathcal{F}_{t-1}\right]$$
$$=\gamma_{t-1}\mathbb{E}\left[\|\bar{\nabla}_{\mu_t}f_{i_t}(x_t)-\bar{\nabla}_{\mu_{t-1}}f_{i_t}(x_{t-1})-\bar{\nabla}_{\mu_t}f(x_t)+\bar{\nabla}_{\mu_{t-1}}f(x_{t-1})\|^2\mid\mathcal{F}_{t-1}\right]$$
$$\quad+\gamma_{t-1}\mathbb{E}\left[\|v_{t-1}-\bar{\nabla}_{\mu_{t-1}}f(x_{t-1})\|^2\mid\mathcal{F}_{t-1}\right]$$
$$=\gamma_{t-1}\mathbb{E}\left[\|\bar{\nabla}_{\mu_t}f_{i_t}(x_t)-\bar{\nabla}_{\mu_{t-1}}f_{i_t}(x_{t-1})\|^2\mid\mathcal{F}_{t-1}\right]+\gamma_{t-1}\mathbb{E}\left[\|v_{t-1}-\bar{\nabla}_{\mu_{t-1}}f(x_{t-1})\|^2\mid\mathcal{F}_{t-1}\right]$$
$$\leq2L^2\gamma_{t-1}\mathbb{E}\left[\|x_t-x_{t-1}\|^2\mid\mathcal{F}_{t-1}\right]+\gamma_{t-1}\mathbb{E}\left[\|v_{t-1}-\bar{\nabla}_{\mu_{t-1}}f(x_{t-1})\|^2\mid\mathcal{F}_{t-1}\right]+4\gamma_{t-1}(\delta_t^2+\delta_{t-1}^2)$$
$$=2L^2\gamma_{t-1}^3\mathbb{E}\left[\|v_{t-1}\|^2\mid\mathcal{F}_{t-1}\right]+\gamma_{t-1}\mathbb{E}\left[\|v_{t-1}-\bar{\nabla}_{\mu_{t-1}}f(x_{t-1})\|^2\mid\mathcal{F}_{t-1}\right]+4\gamma_{t-1}(\delta_t^2+\delta_{t-1}^2).$$

We obtain the following by first conditioning on all randomness up to round $t$, and then taking the total expectation:

$$\mathbb{E}\left[\gamma_t\|v_t-\bar{\nabla}_{\mu_t}f(x_t)\|^2\right]\leq2L^2\mathbb{E}\left[\gamma_{t-1}^3\|v_{t-1}\|^2\right]+\mathbb{E}\left[\gamma_{t-1}\|v_{t-1}-\bar{\nabla}_{\mu_{t-1}}f(x_{t-1})\|^2+4\gamma_{t-1}(\delta_t^2+\delta_{t-1}^2)\right].$$

Since $\mathbb{E}\left[\|v_t-\bar{\nabla}_{\mu_t}f(x_t)\|\right]=0$ whenever $t\bmod n=0$, it follows that

$$\mathbb{E}\left[\gamma_t\cdot\|v_t-\bar{\nabla}_{\mu_t}f(x_t)\|^2\right]\leq2\mathbb{E}\left[\sum_{s=t-t\bmod n}^{t-1}L^2\gamma_s^3\|v_s\|^2+4\gamma_s(\delta_s^2+\delta_{s-1}^2)\right],$$

which leads to:

$$\mathbb{E}\left[\sum_{t=0}^{T-1}\gamma_t\cdot\|v_t-\bar{\nabla}_{\mu_t}f(x_t)\|^2\right]\leq2\mathbb{E}\left[\sum_{t=0}^{T-1}L^2n\gamma_t^3\|v_t\|^2+8n\gamma_t\delta_t^2\right],$$

observe that the first term can be bounded by the following terms:

$$\mathbb{E}\left[\sum_{t=0}^{T-1}\gamma_t^3\|v_t\|^2\right]$$
$$=\frac{1}{c^3}\mathbb{E}\left[\sum_{t=0}^{T-1}\frac{1}{n^{1/2}+\sum_{s=0}^t\|v_t\|^2}\cdot\frac{\|v_t\|^2}{n^{1/2}+\sum_{s=0}^t\|v_s\|^2}\right]$$
$$\leq\frac{1}{c^3n}\mathbb{E}\left[\sum_{t=0}^{T-1}\frac{1}{n^{3/4}\sqrt{n^{1/2}}}\cdot\frac{\|v_t\|^2}{n^{1/2}+\sum_{s=0}^t\|v_s\|^2}\right]$$
$$\leq\frac{1}{c^3n}\cdot\mathbb{E}\left[\sum_{t=0}^{T-1}\frac{\|v_t\|^2}{1+\sum_{s=0}^t\|v_s\|^2}\right]$$
$$\leq\frac{1}{c^3n}\log(\Phi(T)),$$

where the fourth inequality follows by Lemma B.7 and Lemma C.2.

Finally we obtain

$$\mathbb{E}\left[\sum_{t=0}^{T-1}\gamma_t\cdot\|v_t-\bar{\nabla}_{\mu_t}f(x_t)\|^2\right]\leq\frac{2L^2}{c^3}\log(\Phi(T))+\sum_{t=0}^{T-1}16n\gamma_t\delta_t^2.$$

$\square$

## C.2 PART II

**Lemma C.5.** *Under assumptions 1 and 2, we can derive the following result for Algorithm 1:*

$$\frac{1}{T}\mathbb{E}\left[\sum_{t=0}^{T-1}\|v_t - \bar{\nabla}_{\mu_t}f(x_t)\|\right] \leq \frac{Ln^{1/4}}{c\sqrt{T}}\log\left(\Phi(T)\right) + \frac{1}{\sqrt{T}}\sqrt{8n\sum_{t=0}^{T-1}\delta_t^2}.$$

*Proof.*

$$\mathbb{E}\left[\sum_{t=0}^{T-1}\|v_t - \bar{\nabla}_{\mu_t}f(x_t)\|\right] \leq \sqrt{T}\cdot\sqrt{\mathbb{E}\left[\sum_{t=0}^{T-1}\|v_t - \bar{\nabla}_{\mu_t}f(x_t)\|^2\right]},$$

the final inequality holds since $\left\|\sum_{t=0}^{T-1}a_t\right\|^2$ can be bounded by $T\cdot\sum_{t=0}^{T-1}\|a_t\|^2$ using Jensen's inequality . By an argument entirely analogous to that of Lemma C.4, we can establish the same result for the estimator $v_t = \bar{\nabla}_{\mu_t}f_{i_t}(x_t) - \bar{\nabla}_{\mu_{t-1}}f_{i_t}(x_{t-1}) + v_{t-1}$:

$$\mathbb{E}\left[\|v_t - \bar{\nabla}_{\mu_t}f(x_t)\|^2\right] \leq 4(\delta_t^2 + \delta_{t-1}^2) + 2L^2\mathbb{E}\left[\|x_t - x_{t-1}\|^2\right] + \mathbb{E}\left[\|v_{t-1} - \bar{\nabla}_{\mu_{t-1}}f(x_{t-1})\|^2\right]$$

$$\leq L^2\mathbb{E}\left[\gamma_{t-1}^2\|v_{t-1}\|^2\right] + \mathbb{E}\left[\|v_{t-1} - \bar{\nabla}f(x_{t-1})\|^2\right] + 4(\delta_t^2 + \delta_{t-1}^2)$$

$$= \sum_{\tau=t-(t\bmod n)+1}^{t-1} L^2\mathbb{E}\left[\gamma_\tau^2\|v_\tau\|^2\right] + 4(\delta_\tau^2 + \delta_{\tau-1}^2),$$

by a telescoping summation over $t$ we get that

$$\sum_{t=0}^{T-1}\mathbb{E}\left[\|v_t - \bar{\nabla}_{\mu_t}f(x_t)\|^2\right] \leq L^2 n\cdot\mathbb{E}\left[\sum_{t=0}^{T-1}\gamma_t^2\|v_t\|^2\right] + 8n\sum_{t=0}^{T-1}\delta_t^2.$$

Now as discussed in Lemma C.4, using the step-size selection $\gamma_t$ we obatain:

$$\sum_{t=0}^{T-1}\mathbb{E}\left[\|v_t - \bar{\nabla}_{\mu_t}f(x_t)\|^2\right] \leq L^2 n\cdot\mathbb{E}\left[\sum_{t=0}^{T-1}\gamma_t^2\|v_t\|^2\right] + 8n\sum_{t=0}^{T-1}2\delta_t^2$$

$$= \frac{L^2\sqrt{n}}{c^2}\cdot\mathbb{E}\left[\sum_{t=0}^{T-1}\frac{\|v_t\|^2}{\sqrt{n} + \sum_{s=0}^{t}\|\bar{\nabla}_s\|^2}\right] + 8n\sum_{t=0}^{T-1}\delta_t^2$$

$$\leq \frac{L^2\sqrt{n}}{c^2}\log\left(\Phi(T)\right) + 8n\sum_{t=0}^{T-1}\delta_t^2$$

$$\leq \frac{L^2\sqrt{n}}{c^2}\log\left(\Phi(T)\right) + 8n\sum_{t=0}^{T-1}2\delta_t^2,$$

where lstinequality follows by Lemma B.6 and Lemma C.2. Putting everything together we get

$$\frac{1}{T}\mathbb{E}\left[\sum_{t=0}^{T-1}\|v_t - \bar{\nabla}_{\mu_t}f(x_t)\|\right] \leq \frac{Ln^{1/4}}{c\sqrt{T}}\log\left(\Phi(T)\right) + \frac{1}{\sqrt{T}}\sqrt{8n\sum_{t=0}^{T-1}\delta_t^2}.$$

$\square$

## C.3 PART III

$\frac{1}{T}\sum_{t=0}^{T-1}\|\nabla f(x_t) - \bar{\nabla}_{\mu_t}f(x_t)\| \leq \frac{1}{T}\sum_{t=0}^{T-1}\delta_t.$

## C.4 Final proof for coordinate estimator

**Theorem C.1.** *Under assumptions 1 and 2, based on the previous lemmas C.3, C.4, C.5, we can derive the following result for Algorithm 1:*

$$\frac{1}{T}\mathbb{E}[\sum_{t=0}^{T-1}\|\nabla f(x_t)\|] \leq \frac{n^{1/4}}{\sqrt{T}}\left(2\Delta \cdot c + 1 + (\frac{L}{c} + \frac{L^2}{c^2})\log(\Phi(T))\right) + \frac{L^2\pi^2}{24n^{\frac{1}{4}}} + \sqrt{\frac{\pi^2}{24}}\frac{L}{n^{\frac{1}{4}}} + \frac{L^2\pi^2}{12} + \frac{L}{2}\right)$$

*setting $c = 1$, we obtain $T = \tilde{\mathcal{O}}(\sqrt{n}\epsilon^{-2})$, where the $\tilde{\mathcal{O}}$ notation hides logarithmic factors.*

*Proof.*

$$\frac{1}{T}\mathbb{E}[\sum_{t=0}^{T-1}\|\nabla f(x_t)\|] \leq \frac{1}{T}[\sum_{t=0}^{T-1}\mathbb{E}[\|v_t\|] + \sum_{t=0}^{T-1}\mathbb{E}[\|v_t - \bar{\nabla}_{\mu_t}f(x_t)\|] + \|\bar{\nabla}_{\mu_t}f(x_t) - \nabla f(x_t)\|]$$

$$\leq \frac{n^{1/4}}{\sqrt{T}}\left(2\Delta \cdot c + 1 + (\frac{L}{cn^{\frac{3}{4}}} + \frac{L^2}{c^2})\log(\Phi(T))\right)$$

$$+ \frac{1}{T}(2c\sum_{t=0}^{T-1}\gamma_t\delta_t^2 + \sum_{t=0}^{T-1}\delta_t) + \frac{1}{\sqrt{T}}\left(n^{\frac{5}{4}}\sum_{t=0}^{T-1}c\gamma_t\delta_t^2 + n^{\frac{1}{2}}\sqrt{\sum_{t=0}^{T-1}2\delta_t^2}\right).$$

Due to the fact that $\gamma_t \leq \frac{1}{cn^{\frac{1}{4}}}$ we obtain:

$$\frac{1}{T}\mathbb{E}[\sum_{t=0}^{T-1}\|\nabla f(x_t)\|] \leq \frac{n^{1/4}}{\sqrt{T}}\left(2\Delta \cdot c + 1 + (\frac{L}{cn^{\frac{3}{4}}} + \frac{L^2}{c^2})\log(\Phi(T))\right)$$

$$+ \frac{1}{T}(2\sum_{t=0}^{T-1}\frac{1}{n^{\frac{1}{4}}}\delta_t^2 + \sum_{t=0}^{T-1}\delta_t) + \frac{1}{\sqrt{T}}\left(n\sum_{t=0}^{T-1}\delta_t^2 + n^{\frac{1}{2}}\sqrt{\sum_{t=0}^{T-1}2\delta_t^2}\right).$$

Take $\delta_t = \frac{L}{2\sqrt{n}(t+1)}$ i.e.$(\mu_t = \frac{1}{\sqrt{nd}}(t+1))$ then :

$$\sum_{t=0}^{T-1}\delta_t \leq \frac{L ln T}{\sqrt{n}}.$$

$$\sum_{t=0}^{T-1}\delta_t^2 < \frac{L^2\pi^2}{24n}.$$

combing the above results we obtain:

$$\frac{1}{T}\mathbb{E}[\sum_{t=0}^{T-1}\|\nabla f(x_t)\|] \leq \frac{n^{1/4}}{\sqrt{T}}\left(2\Delta \cdot c + 1 + (\frac{L}{cn^{\frac{3}{4}}} + \frac{L^2}{c^2})\log(\Phi(T))\right) + \frac{1}{T}(\frac{L^2\pi^2}{12} + \frac{L ln T}{2}) + \frac{1}{\sqrt{T}}\left(\frac{L^2\pi^2}{24} + L\sqrt{\frac{\pi^2}{24}}\right)$$

$$\leq \frac{n^{1/4}}{\sqrt{T}}\left(2\Delta \cdot c + 1 + (\frac{L}{cn^{\frac{3}{4}}} + \frac{L^2}{c^2})\log(\Phi(T)) + \frac{L^2\pi^2}{24n^{\frac{1}{4}}} + \sqrt{\frac{\pi^2}{24}}\frac{L}{n^{\frac{1}{4}}} + \frac{L^2\pi^2}{12} + \frac{L}{2}\right),$$

setting $c = 1$, we obtain $T = \tilde{\mathcal{O}}(\sqrt{n}\epsilon^{-2})$, where the $\tilde{\mathcal{O}}$ notation hides logarithmic factors. $\qquad\square$

# D PARAMETER FREE VARIANCE REDUCED ZEROTH-ORDER METHOD(RANDOM-DIRECTION ESTIMATOR)

---

**Algorithm 4** PF-VRZO(Random-direction)

---

Set $c = \sqrt{d}$ for random-direction estimator and $\mu_{-1} = \mu_0$.
**for** $t = 0$ **to** $T-1$ **do**
    Compute smoothing parameter $\mu_t = \frac{1}{(t+1)d\sqrt{n}}$ , smoothing vector $\rho_t \sim U_B$.
    **if** $t \bmod n = 0$ **then**
        $v_t = \hat{\nabla}_{\mu_t} f(x_t)$ {Full zeroth-order gradient computation}
    **else**
        Sample $i_t \in \{1, \ldots, n\}$ uniformly at random
        Compute $\hat{\nabla}_{\mu_t} f_{i_t}(x_t)$ with parameter $\mu_t$ and rand vector $\rho_t$, $\hat{\nabla}_{\mu_{t-1}} f_{i_t}(x_{t-1})$ with different
        parameter $\mu_{t-1}$ and the same rand vector $\rho_t$.
        $v_t = \hat{\nabla}_{\mu_t} f_{i_t}(x_t) - \hat{\nabla}_{\mu_{t-1}} f_{i_t}(x_{t-1}) + v_{t-1}$
    **end if**
    $\gamma_t = \dfrac{1}{n^{1/4}c\sqrt{(n^{1/2} + \sum_{s=0}^t \|v_s\|^2)}}$
    $x_{t+1} = x_t - \gamma_t v_t$
**end for**

---

Table 3: Meaning of Symbols

| Symbol | Meaning |
|--------|---------|
| $\gamma_t$ | stepsize $\frac{1}{\left(n^{1/4}c\sqrt{n^{1/2}+\sum_{s=0}^t \|v_s\|^2}\right)}$ . |
| $\mu_t$ | Smoothing parameter at iteration $t$. |
| $\rho_t$ | Smoothing vector at iteration $t$. |
| $v_t$ | Spider operator. |
| $\nabla_{\mu_t} f(x_t)$ | zeroth-order estimator(rand) using smoothing parameter $\mu_t$ and $\rho_t$ . |
| $\hat{\nabla} f_\mu(\cdot)$ | expecation of zeroth-order estimator(rand). |
| $\Delta_t$ | $Ld\mu_t/2$, the estimation error with respect to $\nabla f_\mu(\cdot)$ . |

Following a similar approach as with the coordinate operator, we analyze the convergence of the gradient of $f(x)$ by dividing it into three parts.

$$\frac{1}{T}\mathbb{E}[\sum_{t=0}^{T-1} \|\nabla f(x_t)\|] \leq \frac{1}{T}[\underbrace{\sum_{t=0}^{T-1} \mathbb{E}[\|v_t\|]}_{\text{part I}} + \underbrace{\sum_{t=0}^{T-1} \mathbb{E}[\|v_t - \nabla f_{\mu_t}(x)\|]}_{\text{part II}} + \underbrace{\sum_{t=0}^{T-1} \|\nabla f_{\mu_t}(x) - \nabla f(x_t)\|}_{\text{part III}}].$$

**Lemma D.1** ((Ji et al., 2019)). *Let $f_\mu(x) = \mathbb{E}_{w \sim U_B}[f(x + \mu w)]$ be a smooth approximation of $f(x)$, where $U_B$ is the uniform distribution over the d-dimensional unit Euclidean ball B, and $\rho \in \mathbb{R}^d$ is a random vector sampled from unit Euclidean sphere $U_{S_p}$. Then we have*

1. *$|f_\mu(x) - f(x)| \leq \frac{\mu^2 L}{2}$ and $\|\nabla f_\mu(x) - \nabla f(x)\| \leq \frac{\mu L d}{2}$ for any $x \in \mathbb{R}^d$.*

2. *$\mathbb{E}\|\hat{\nabla}_\mu f_i(x_1) - \hat{\nabla}_\mu f_i(x_2)\|^2 \leq 3dL^2\|x_1 - x_2\|^2 + \frac{3L^2 d^2 \mu^2}{2}$ for any $i$ and any $x_1, x_2 \in \mathbb{R}^d$.*

3. *$\mathbb{E}_{\rho \sim U_{S_p}}\left[\|\hat{\nabla} f(x)\|^2\right] \leq 2d\|\nabla f(x)\|^2 + \frac{L^2 \mu^2 d^2}{2}$.*

**Lemma D.2.** *For random-direction estimator* $\hat{\nabla}_{\mu_t} f_i(x_t) = \frac{d}{\mu_t}[f(x_t + \mu_t\rho_t) - f(x_t)]\rho_t, \hat{\nabla}_{\mu_{t-1}} f_i(x_{t-1}) = \frac{d}{\mu_{t-1}}[f(x + \mu_{t-1}\rho_t) - f(x_{t-1})]\rho_t$, *where both estimators use the same random direction* $\rho_t$ *sapled from unit Euclidean sphere* $U_{S_p}$ *but different smoothing parameters* $\mu_t$ *and* $\mu_{t-1}$, *we have:*

$$\|\hat{\nabla}_{\mu_t} f(x_t) - \hat{\nabla}_{\mu_{t-1}} f(x_{t-1})\|^2 \le \frac{3}{2}(\Delta_t^2 + \Delta_{t-1}^2) + 3dL^2\|x_t - x_{t-1}\|^2.$$

*Proof.*

$$\mathbb{E}\left[\|\hat{\nabla}_{\mu_t} f(x_t) - \hat{\nabla}_{\mu_{t-1}} f(x_t)\|^2\right]$$

$$= d^2\mathbb{E}\left[\left\|\frac{\rho_t}{\mu_t}\Big[f(x_t + \mu_t\rho_t) - f(x_t) - \langle\nabla f(x_t), \rho_t\rangle\Big]\rho_t - \frac{\rho_t}{\mu_{t-1}}\Big[f(x_{t-1} + \mu_{t-1}\rho_t) - f(x_{t-1}) - \langle\nabla f(x_{t-1}), \rho_t\rangle\Big]\rho_t\right.\right.$$

$$\left.\left. + \Big(\langle\nabla f(x_t), \rho_t\rangle\rho_t - \langle\nabla f(x_{t-1}), \rho_t\rangle\rho_t\Big)\right\|^2\right]$$

$$\le d^2\left(\frac{3L^2}{2}(\mu_t^2 + \mu_{t-1}^2) + \mathbb{E}\left[3\|\langle\nabla f(x_t), \rho_t\rangle\rho_t - \langle\nabla f(x_{t-1}), \rho_t\rangle\rho_t\|^2\right]\right)$$

$$= d^2\left(\frac{3L^2}{2}(\mu_t^2 + \mu_{t-1}^2) + \mathbb{E}\left[3\|\langle\nabla f(x_{t-1}) - \nabla f(x_t), \rho_t\rangle\|^2\right]\right) \qquad (\|\rho_t\|^2 = 1)$$

$$\le d^2\left(\frac{3L^2}{2}(\mu_t^2 + \mu_{t-1}^2) + \mathbb{E}\left[\frac{3}{d}\|\nabla f(x_{t-1}) - \nabla f(x_t)\|^2\right]\right) \qquad (\mathbb{E}[\rho_t\rho_t^T] = \frac{1}{d}I_d \text{ (Ji et al., 2019)})$$

$$\le \frac{3}{2}(\Delta_t^2 + \Delta_{t-1}^2) + 3dL^2\|x_t - x_{t-1}\|^2.$$

□

**Lemma D.3.** *Under assumptions 1 and 2, we can derive the following result for Algorithm 2*

$$\mathbb{E}[\sum_{t=0}^{T-1}\|v_t\|^2] \le \phi(T) + 1.$$

*where* $\phi(T) := \frac{6dL^2 n^{1.5}}{c^2}T + \frac{4dL^2 T^3}{n}c^2 + 4dnT\|\nabla f(x_0)\|^2 + (6n^2 + 2)\sum_{t=0}^{T-1}\Delta_t^2 - 1$. *Similar to the coordwise method, the notation* $\phi(T)$ *is introduced only for brevity, and will be repeatedly used in the subsequent analysis.*

*Proof.*

$$\mathbb{E}\|v_t\|^2 = \left\|\sum_{s=t-t \bmod n+1}^{t}\left(\hat{\nabla}_{\mu_s} f_{i_s}(x_s) - \hat{\nabla}_{\mu_{s-1}} f_{i_s}(x_{s-1})\right) + \hat{\nabla}_{\mu_{t-t \bmod n}} f(x_{t-t \bmod n})\right\|^2$$

$$\le 2\mathbb{E}\left\|\sum_{s=t-t \bmod n+1}^{t}\hat{\nabla}_{\mu_s} f_{i_s}(x_s) - \hat{\nabla}_{\mu_{s-1}} f_{i_s}(x_{s-1})\right\|^2 + 2\mathbb{E}\left\|\hat{\nabla}_{\mu_{t-t \bmod n}} f(x_{t-t \bmod n})\right\|^2$$

$$\le 2n\mathbb{E}\sum_{s=t-t \bmod n+1}^{t}\left\|\hat{\nabla}_{\mu_s} f_{i_s}(x_s) - \hat{\nabla}_{\mu_{s-1}} f_{i_s}(x_{s-1})\right\|^2 + 2\mathbb{E}\left\|\hat{\nabla}_{\mu_{t-t \bmod n}} f(x_{t-t \bmod n})\right\|^2$$

$$\overset{lem D.2}{\le} 6n\sum_{s=t-t \bmod n+1}^{t}[dL^2(x_s - x_{s-1})^2 + \frac{1}{2}(\Delta_s^2 + \Delta_{s-1}^2)] + 2\mathbb{E}\left\|\hat{\nabla}_{\mu_{t-t \bmod n}} f(x_{t-t \bmod n})\right\|^2$$

$$\le \frac{6dL^2 n^2}{c^2} + 3n\sum_{s=t-t \bmod n+1}^{t}(\Delta_s^2 + \Delta_{s-1}^2) + 2\mathbb{E}\left\|\hat{\nabla}_{\mu_{t-t \bmod n}} f(x_{t-t \bmod n})\right\|^2,$$

from lemma D.1:

$$\mathbb{E}_{\rho\sim U_{S_p}}\left[\|\hat{\nabla}_{\mu_t} f(x)\|^2\right] \le 2d\|\nabla f(x)\|^2 + \Delta_t^2.$$

Next, we bound $\|\nabla f(x)\|$ bellow:

$$
\begin{aligned}
\|\nabla f(x_t)\| &= \|\nabla f(x_t) - \nabla f(x_0) + \nabla f(x_0)\| \\
&\leq \|\nabla f(x_t) - \nabla f(x_0)\| + \|\nabla f(x_0)\| \\
&\leq L\|x_t - x_0\| + \|\nabla f(x_0)\| \\
&\leq L\|x_t - x_{t-1}\| + L\|x_{t-1} - x_0\| + \|\nabla f(x_0)\| \\
&\leq L\sum_{i=1}^{t} \|x_i - x_{i-1}\| + \|\nabla f(x_0)\| \\
&\leq \frac{Lt}{c\sqrt{n}} + \|\nabla f(x_0)\|.
\end{aligned}
$$

Combine the above results, we have:

$$
\begin{aligned}
\mathbb{E}[\sum_{t=0}^{T-1} \|v_t\|^2] &\leq \sum_{t=0}^{T-1} \left( \frac{6L^2 n}{c^2} + 3n \sum_{s=t-t \bmod n+1}^{t} (\Delta_s^2 + \Delta_{s-1}^2) + \mathbb{E}[2\left\|\hat{\nabla} f_\mu \left(x_{t-t \bmod n}\right)\right\|^2] \right) \\
&\leq \frac{6dTL^2 n}{c^2} + 2d\sum_{t=0}^{T-1} \|\nabla f(x_t)\|^2 + 3n\sum_{t=0}^{T-1}\sum_{s=t-t \bmod n+1}^{t} (\Delta_s^2 + \Delta_{s-1}^2) + 2\sum_{t=0}^{T-1} \Delta_t^2 \\
&\leq \frac{6L^2 dn}{c^2}T + 2d\sum_{t=0}^{T-1} \left( \frac{Lt}{\sqrt{n}c} + \|\nabla f(x_0)\| \right)^2 + (6n^2 + 2)\sum_{t=0}^{T-1} \Delta_t^2 \\
&\leq \frac{6dL^2 n}{c^2}T + \frac{4dL^2 T^3}{n}c^2 + 4dnT\|\nabla f(x_0)\|^2 + (6n^2 + 2)\sum_{t=0}^{T-1} \Delta_t^2.
\end{aligned}
$$

Similar to the coordwise method, we define $\phi(T) := \frac{6dL^2 n^{1.5}}{c^2}T + \frac{4dL^2 T^3}{n}c^2 + 4dnT\|\nabla f(x_0)\|^2 + (6n^2 + 2)\sum_{t=0}^{T-1}\Delta_t^2 - 1$ to simplify the resulting expressions. $\qquad\square$

## D.1 PART I

**Lemma D.4** (part I(1)). *Under assumptions 1 and 2, we can derive the following result for Algorithm 2*

$$
\mathbb{E}\left[\sum_{t=0}^{T-1}\|v_t\|\right] \leq n^{1/4}\sqrt{T}\left(2\Delta c + 1 + \frac{L}{c}\log(\phi(T)) + c\cdot\mathbb{E}\left[\sum_{t=0}^{T-1}\gamma_t\|\nabla f(x_t) - v_t\|^2\right]\right)
$$

*Proof.*

$$
\begin{aligned}
\mathbb{E}\left[f(x_{t+1}) \mid \mathcal{F}_t\right] &\leq \mathbb{E}\left[f(x_t) + \nabla f(x_t)^T(x_{t+1} - x_t) + \frac{L}{2}\|x_t - x_{t+1}\|^2 \mid \mathcal{F}_t\right] \\
&= \mathbb{E}\left[f(x_t) - \gamma_t v_t^T \nabla f(x_t) + \frac{L}{2}\gamma_t^2\|v_t\|^2 \mid \mathcal{F}_t\right] \\
&\leq \mathbb{E}\left[f(x_t) + 2\gamma_t\|v_t - \nabla f(x_t)\|^2 - \frac{\gamma_t}{2}(1 - L\gamma_t)\|v_t\|^2 \mid \mathcal{F}_t\right],
\end{aligned}
$$

which leads to:

$$
\mathbb{E}\left[\gamma_t \cdot \|v_t\|^2\right] \leq 2\mathbb{E}\left[f(x_t) - f(x_{t+1})\right] + \mathbb{E}\left[L\gamma_t^2 \cdot \|v_t\|^2\right] + 2c\cdot\mathbb{E}\left[\gamma_t \cdot \|\nabla f(x_t) - v_t\|^2\right].
$$

By summing from $t = 0$ to $T - 1$ we get:

$$
\sum_{t=0}^{T-1}\mathbb{E}\left[\gamma_t \cdot \|v_t\|^2\right] \leq 2\Delta + \mathbb{E}\left[\sum_{t=0}^{T-1} L\gamma_t^2 \cdot \|v_t\|^2\right] + \mathbb{E}\left[\sum_{t=0}^{T-1} 2\gamma_t \cdot \|\nabla f(x_t) - v_t\|^2\right].
$$

Recall that $\gamma_t = n^{-1/4}c^{-1}\left(n^{1/2} + \sum_{s=0}^{t}\|v_s\|^2\right)^{-1/2}$:

$$\mathbb{E}\left[\sum_{t=0}^{T-1}\gamma_t \cdot \|v_t\|^2\right] \leq 2\Delta + \frac{L}{c^2}\cdot\mathbb{E}\left[\sum_{t=0}^{T-1}\frac{\|v_t\|^2}{\sqrt{n} + \sum_{s=0}^{t}\|v_s\|^2}\right] + \mathbb{E}\left[\sum_{t=0}^{T-1}\gamma_t\cdot\|\nabla f(x_t) - v_t\|^2\right]$$

$$\leq 2\Delta + \frac{L}{\sqrt{n}c^2}\log(\phi(T)) + \mathbb{E}\left[\sum_{t=0}^{T-1}\gamma_t\cdot\|\nabla f(x_t) - v_t\|^2\right].$$

Lower bounding the right-hand side:

$$\mathbb{E}\left[\sum_{t=0}^{T-1}\gamma_t\cdot\|v_t\|^2\right] \geq \mathbb{E}\left[\frac{\sum_{t=0}^{T-1}\|v_t\|^2}{n^{1/4}c\sqrt{n^{1/2} + \sum_{t=0}^{T-1}\|v_t\|^2}}\right]$$

$$\geq \frac{1}{c}\cdot\mathbb{E}\left[\frac{\sum_{t=0}^{T-1}\|v_t\|^2/\sqrt{n}}{\sqrt{1 + \sum_{t=0}^{T-1}\|v_t\|^2/\sqrt{n}}}\right]$$

$$\geq \frac{1}{c}\cdot\mathbb{E}\left[\sum_{t=0}^{T-1}\|v_t\|^2/\sqrt{n}\right] - 1$$

$$\geq \frac{1}{cn^{1/4}\sqrt{T}}\mathbb{E}\left[\sum_{t=0}^{T-1}\|v_t\|\right] - \frac{1}{c}.$$

Combining all results:

$$\mathbb{E}\left[\sum_{t=0}^{T-1}\|v_t\|\right] \leq n^{1/4}\sqrt{T}\left(2\Delta c + 1 + \frac{L}{c\sqrt{n}}\log(\phi(T)) + c\cdot\mathbb{E}\left[\sum_{t=0}^{T-1}\gamma_t\|\nabla f(x_t) - v_t\|^2\right]\right).$$

$\square$

**Lemma D.5** (part I(2)). *Under assumptions 1 and 2, we can derive the following result for Algorithm 2*

$$\mathbb{E}\left[\sum_{t=0}^{T-1}\gamma_t\cdot\|v_t - \nabla f(x_t)\|^2\right] \leq \frac{6dL^2}{c^3}\log(\phi(T)) + \sum_{t=0}^{T-1}(3n+8)n\gamma_t\Delta_t^2.$$

*Proof.* Let $\mathcal{F}_t$ be the sigma-algebra generated by $\{i_0, \ldots, i_t\}$ and $x_0$. From the definition of $\gamma_t$, it follows that $\gamma_t \leq \gamma_{t-1}$; this condition is imposed to resolve measurability concerns. Consequently,

$$\mathbb{E}\left[\gamma_t\|v_t - \nabla f(x_t)\|^2 \mid \mathcal{F}_{t-1}\right] \leq \mathbb{E}\left[\gamma_{t-1}\cdot\|v_t - \nabla f(x_t)\|^2 \mid \mathcal{F}_{t-1}\right],$$

Hence, our analysis can be reduced to studying $\mathbb{E}\left[\gamma_{t-1}\|v_t - \nabla f(x_t)\|^2 \mid \mathcal{F}_{t-1}\right]$.

$$\mathbb{E}\left[\gamma_{t-1}\|v_t - \nabla f(x_t)\|^2 \mid \mathcal{F}_{t-1}\right]$$
$$\leq 2\mathbb{E}\left[\gamma_{t-1}\|v_t - \nabla f_{\mu_t}(x_t)\|^2 \mid \mathcal{F}_{t-1}\right] + 2\mathbb{E}\left[\gamma_{t-1}\|\nabla f(x_t) - \nabla f_{\mu_t}(x_t)\|^2 \mid \mathcal{F}_{t-1}\right].$$

As established in Lemma D.1, the second term can be bounded by $4\gamma_{t-1}\Delta_t^2$ . In the following, we focus on the analysis of the first term.

$$\mathbb{E}\left[\gamma_{t-1}\|v_t - \nabla f_{\mu_t}(x_t)\|^2 \mid \mathcal{F}_{t-1}\right]$$

$$= \gamma_{t-1}\mathbb{E}\left[\|\hat{\nabla}_{\mu_t}f_{i_t}(x_t) - \hat{\nabla}_{\mu_{t-1}}f_{i_t}(x_{t-1}) - \nabla f_{\mu_t}(x_t) + \nabla f_{\mu_{t-1}}(x_{t-1}) + (v_{t-1} - \nabla f_{\mu_{t-1}}(x_{t-1}))\|^2 \mid \mathcal{F}_{t-1}\right]$$

$$= \gamma_{t-1}\mathbb{E}\left[\|\hat{\nabla}_{\mu_t}f_{i_t}(x_t) - \hat{\nabla}_{\mu_{t-1}}f_{i_t}(x_{t-1}) - \nabla f_{\mu_{t-1}}(x_{t-1}) + (v_{t-1} - \nabla f_{\mu_{t-1}}(x_{t-1})\|^2) \mid \mathcal{F}_{t-1}\right]$$

$$\quad + \gamma_{t-1}\mathbb{E}\left[\|v_{t-1} - \nabla f(x_{t-1})\|^2 \mid \mathcal{F}_{t-1}\right]$$

$$\leq \gamma_{t-1}\mathbb{E}\left[\|\hat{\nabla}_{\mu_t}f_{i_t}(x_t) - \hat{\nabla}_{\mu_{t-1}}f_{i_t}(x_{t-1})\|^2 \mid \mathcal{F}_{t-1}\right] + \gamma_{t-1}\mathbb{E}\left[\|v_{t-1} - \nabla f_{\mu_{t-1}}(x_{t-1})\|^2 \mid \mathcal{F}_{t-1}\right]$$

$$\overset{Lem D.2}{\leq} 3dL^2\gamma_{t-1}\mathbb{E}\left[\|x_t - x_{t-1}\|^2 \mid \mathcal{F}_{t-1}\right] + \gamma_{t-1}\mathbb{E}\left[\|v_{t-1} - \nabla f_{\mu_{t-1}}(x_{t-1})\|^2 \mid \mathcal{F}_{t-1}\right] + \frac{3\gamma_{t-1}}{2}(\Delta_t^2 + \Delta_{t-1}^2)$$

$$= 3dL^2\gamma_{t-1}^3\mathbb{E}\left[\|v_{t-1}\|^2 \mid \mathcal{F}_{t-1}\right] + \gamma_{t-1}\mathbb{E}\left[\|v_{t-1} - \nabla f_{\mu_{t-1}}(x_{t-1})\|^2 \mid \mathcal{F}_{t-1}\right] + \frac{3\gamma_{t-1}}{2}(\Delta_t^2 + \Delta_{t-1}^2).$$

$$(4)$$

We obtain the following by first conditioning on all randomness up to round $t$, and then taking the total expectation:

$$\mathbb{E}\left[\gamma_t\|v_t - \nabla f_{\mu_t}(x_t)\|^2\right] \leq \mathbb{E}\left[\gamma_{t-1}\|v_{t-1} - \nabla f_{\mu_{t-1}}(x_{t-1})\|^2\right] + 3dL^2\mathbb{E}\left[\gamma_{t-1}^3\|v_{t-1}\|^2\right] + \frac{3\gamma_{t-1}}{2}(\Delta_t^2 + \Delta_{t-1}^2).$$

Since $\mathbb{E}\left[\gamma_t \cdot \|v_t - \nabla f_{\mu_t}(x_t)\|^2\right] \leq \gamma_{t-1}\mathbb{E}\left[\|v_t - \nabla f_{\mu_t}(x_t)\|^2\right] = 0$ whenever $t \bmod n = 0$, it follows that

$$\mathbb{E}\left[\gamma_t \cdot \|v_t - \nabla f_{\mu_t}(x_t)\|^2\right] \leq \mathbb{E}\left[\sum_{s=t-t \bmod n}^{t-1} 3dL^2\gamma_s^3\|v_s\|^2 + \frac{3\gamma_s}{2}(\Delta_s^2 + \Delta_{s+1}^2)\right].$$

Combine the above results we obtain:

$$\mathbb{E}\left[\gamma_{t-1}\|v_t - \nabla f(x_t)\|^2 \mid \mathcal{F}_{t-1}\right]$$

$$\overset{lem D.1}{\leq} 2\mathbb{E}\left[\gamma_{t-1}\|v_t - \nabla f_{\mu_t}(x_t)\|^2 \mid \mathcal{F}_{t-1}\right] + 2\mathbb{E}\left[\gamma_{t-1}\|\nabla f(x_t) - \nabla f_{\mu_t}(x_t)\|^2 \mid \mathcal{F}_{t-1}\right]$$

$$\leq \mathbb{E}\left[\sum_{s=t-t \bmod n}^{t-1} 6dL^2\gamma_s^3\|v_s\|^2 + 3\gamma_{s-1}(\Delta_s^2 + \Delta_{s+1}^2)\right] + 8\gamma_{t-1}\Delta_t^2,$$

summing over $t$ from 0 to $t-1$ we get that

$$\mathbb{E}\left[\sum_{t=0}^{T-1}\gamma_t \cdot \|v_t - \nabla f(x_t)\|^2\right] \leq \mathbb{E}\left[\sum_{t=0}^{T-1} 6dL^2 n\gamma_t^3\|v_t\|^2 + (3n+8)\gamma_t\Delta_t^2\right].$$

Observe that the first term can be bounded by the following terms:

$$\mathbb{E}\left[\sum_{t=0}^{T-1}\gamma_t^3\|v_t\|^2\right]$$

$$= \frac{1}{c^3}\mathbb{E}\left[\sum_{t=0}^{T-1}\frac{1}{n^{1/2} + \sum_{s=0}^{t}\|v_s\|^2} \cdot \frac{\|v_t\|^2}{n^{1/2} + \sum_{s=0}^{t}\|v_s\|^2}\right]$$

$$\leq \frac{1}{c^3 n}\mathbb{E}\left[\sum_{t=0}^{T-1}\frac{1}{n^{3/4}\sqrt{n^{1/2}}} \cdot \frac{\|v_t\|^2}{n^{1/2} + \sum_{s=0}^{t}\|v_s\|^2}\right]$$

$$\leq \frac{1}{c^3 n} \cdot \mathbb{E}\left[\sum_{t=0}^{T-1}\frac{\|v_t\|^2}{1 + \sum_{s=0}^{t}\|v_s\|^2}\right]$$

$$\leq \frac{1}{c^3 n}\log(\phi(T)),$$

where the fourth inequality follows by Lemma B.7 and Lemma D.3.

Finally we obtain

$$\mathbb{E}\left[\sum_{t=0}^{T-1}\gamma_t \cdot \|v_t - \nabla f(x_t)\|^2\right] \leq \frac{6dL^2}{c^3}\log(\phi(T)) + \sum_{t=0}^{T-1}(3n+8)n\gamma_t\Delta_t^2.$$

$\square$

### D.2 PART II

**Lemma D.6.** *Under assumptions 1 and 2, we can derive the following result for Algorithm 2*

$$\frac{1}{T}\mathbb{E}\left[\sum_{t=0}^{T-1}\|v_t - \nabla f_{\mu_t}(x_t)\|\right] \leq \frac{6\sqrt{d}Ln^{1/4}}{c\sqrt{T}}\log(\phi(T)) + \frac{1}{\sqrt{T}}\sqrt{(3n+8)\sum_{t=0}^{T-1}2\Delta_t^2}.$$

*Proof.*

$$\mathbb{E}\left[\sum_{t=0}^{T-1}\|v_t - \nabla f_{\mu_t}(x_t)\|\right] \leq \sqrt{T} \cdot \sqrt{\mathbb{E}\left[\sum_{t=0}^{T-1}\|v_t - \nabla f_{\mu_t}(x_t))\|^2\right]},$$

where the inequality follows by the fact that $\|\sum_{t=0}^{T-1}y_t\|^2 \leq T \cdot \sum_{t=0}^{T-1}\|y_t\|^2$. For the same reason with equation 4, we obtain:

$$\mathbb{E}\left[\|v_t - \nabla f_{\mu_t}(x_t)\|^2\right] \leq \mathbb{E}\left[\sum_{s=t-t \mod n}^{t-1}6dL^2\gamma_s^2\|v_s\|^2 + 3(\Delta_s^2 + \Delta_{s+1}^2)\right] + 8\Delta_t^2,$$

by a telescoping summation over $t$ we get that

$$\sum_{t=0}^{T-1}\mathbb{E}\left[\|v_t - \nabla f_{\mu_t}(x_t)\|^2\right] \leq \mathbb{E}\left[\sum_{t=0}^{T-1}6dL^2n\gamma_t^2\|v_t\|^2 + (3n+8)\Delta_t^2\right].$$

Using the step-size selection $\gamma_t$ we can provide a bound on the total variance $\mathbb{E}\left[\|v_t - \nabla f_{\mu_t}(x_t)\|^2\right]$:

$$\sum_{t=0}^{T-1}\mathbb{E}\left[\|v_t - \nabla f_{\mu_t}(x_t)\|^2\right]$$

$$\leq 6dL^2n \cdot \mathbb{E}\left[\sum_{t=0}^{T-1}\gamma_t^2\|v_t\|^2\right] + (3n+8)\sum_{t=0}^{T-1}2\Delta_t^2$$

$$= \frac{6dL^2\sqrt{n}}{c^2} \cdot \mathbb{E}\left[\sum_{t=0}^{T-1}\frac{\|v_t\|^2}{\sqrt{n} + \sum_{s=0}^{t}\|v_s\|^2}\right] + (3n+8)\sum_{t=0}^{T-1}\Delta_t^2$$

$$\leq \frac{6dL^2\sqrt{n}}{c^2}\log\left(1 + \mathbb{E}\left[\sum_{t=0}^{T-1}\|v_t\|^2\right]\right) + (3n+8)\sum_{t=0}^{T-1}\Delta_t^2$$

$$\leq \frac{6dL^2\sqrt{n}}{c^2}\log(\phi(T)) + (3n+8)\sum_{t=0}^{T-1}2\Delta_t^2,$$

where last inequality follows by Lemma B.7 and Lemma D.3. Putting everything together we get

$$\frac{1}{T}\mathbb{E}\left[\sum_{t=0}^{T-1}\|v_t - \nabla f_{\mu_t}(x_t)\|\right] \leq \frac{6\sqrt{d}Ln^{1/4}}{c\sqrt{T}}\log(\phi(T)) + \frac{1}{\sqrt{T}}\sqrt{(3n+8)\sum_{t=0}^{T-1}2\Delta_t^2}.$$

$\square$

### D.3 PART III

$\sum_{t=0}^{T-1} \|\nabla f_{\mu_t}(x) - \nabla f(x_t)\| \leq \frac{1}{T} \sum_{t=0}^{T-1} \Delta_t$

### D.4 FINAL PROOF FOR THE RANDOM-DIRECTION ESTIMATOR

**Theorem D.1.** *Under assumptions 1 and 2, based on the previous lemmas D.4, D.5, D.6, we can derive the following result for Algorithm 2:*

$$\frac{1}{T}\mathbb{E}[\sum_{t=0}^{T-1} \|\nabla f(x_t)\|] \leq \frac{n^{1/4}}{\sqrt{T}}\left(\Delta \cdot c + 1 + (\frac{L\sqrt{d}}{cn^{\frac{3}{4}}} + \frac{L^2 d}{c^2})\log(\phi(T)) + \frac{L^2\pi^2}{24n^{\frac{1}{4}}} + \frac{L}{n^{\frac{1}{4}}}\sqrt{\frac{\pi^2}{24}} + \frac{L^2\pi^2}{12} + \frac{L}{2}\right),$$

*setting $c = \sqrt{d}$, we obtain $T = \tilde{\mathcal{O}}(d\sqrt{n}\epsilon^{-2})$, where the $\tilde{\mathcal{O}}$ notation hides logarithmic factors.*

*Proof.*

$$\frac{1}{T}\mathbb{E}[\sum_{t=0}^{T-1} \|\nabla f(x_t)\|] \leq \frac{1}{T}[\underbrace{\sum_{t=0}^{T-1}\mathbb{E}[\|v_t\|]}_{\text{part I}} + \underbrace{\sum_{t=0}^{T-1}\mathbb{E}[\|v_t - \nabla f_{\mu_t}(x)\|]}_{\text{part II}} + \underbrace{\sum_{t=0}^{T-1} \|\nabla f_{\mu_t}(x) - \nabla f(x_t)\|]}_{\text{part III}}$$

$$\leq \frac{n^{1/4}}{\sqrt{T}}\left(2\Delta \cdot c + 1 + (\frac{L\sqrt{d}}{cn^{\frac{3}{4}}} + \frac{dL^2}{c^2})\log(\phi(T))\right)$$

$$+ \frac{1}{T}(2c\sum_{t=0}^{T-1}\gamma_t\Delta_t^2 + \sum_{t=0}^{T-1}\Delta_t) + \frac{1}{\sqrt{T}}\left(n^{\frac{5}{4}}\sum_{t=0}^{T-1}c\gamma_t\Delta_t^2 + n^{\frac{1}{2}}\sqrt{\sum_{t=0}^{T-1} 2\Delta_t^2}\right),$$

take $\Delta_t = \frac{L}{2\sqrt{n}(t+1)}$ i.e.$(\mu_t = \frac{1}{d\sqrt{n}(t+1)})$ , from we obtain:

$$\sum_{t=0}^{T-1}\Delta_t \leq \frac{L ln T}{2\sqrt{n}}.$$

$$\sum_{t=0}^{T-1}\Delta_t^2 < \frac{L^2\pi^2}{24n}.$$

combing the above results we obtain:

$$\frac{1}{T}\mathbb{E}[\sum_{t=0}^{T-1} \|\nabla f(x_t)\|] \leq \frac{n^{1/4}}{\sqrt{T}}\left(\Delta \cdot c + 1 + (\frac{L\sqrt{d}}{cn^{\frac{3}{4}}} + \frac{L^2 d}{c^2})\log(\phi(T)) + \frac{L^2\pi^2}{24n^{\frac{1}{4}}} + \frac{L}{n^{\frac{1}{4}}}\sqrt{\frac{\pi^2}{24}} + \frac{L^2\pi^2}{12} + \frac{L}{2}\right),$$

setting $c = \sqrt{d}$, we obtain $T = \tilde{\mathcal{O}}(d\sqrt{n}\epsilon^{-2})$, where the $\tilde{\mathcal{O}}$ notation hides logarithmic factors. $\qquad\square$

## E HYPERPARAMETERS DETAILS

### E.1 PHASE RETRIEVAL

We choose the problem dimension to be $d = 100$ and the sample size to be $n = 3000$. The measurement vectors $a_r \in \mathbb{R}^d$ and the true parameter $z \in \mathbb{R}^d$ are generated element-wise from a Gaussian distribution $\mathcal{N}(0, 0.5)$. For the initialization, $z_0 \in \mathbb{R}^d$ is drawn element-wise from $\mathcal{N}(5, 0.5)$. The measurements are then constructed as $y_i = |a_r^T z|^2 + m_i$ for $i = 1, \ldots, n$, where the noise term $m_i$ is sampled from $\mathcal{N}(0, 4^2)$, representing additive Gaussian noise.

We set the parameters for ZO-SGD with a learning rate of $\gamma = 2 \times 10^{-8}$ and a batch size of $\sqrt{n}$. For ZO-SPIDER-coord and ZO-SPIDER-rand, we set the learning rate to $\gamma = 10^{-7}$, the epoch size to $q = n$, and the batch sizes to $B = n$ and $B' = 1$. For the proposed PF-VRZO method, we similarly set the epoch size to $q = n$, and choose $B = n$ and $B' = 1$ for both the coord and random-direction estimators.

## E.2 DRO

We set the parameters for ZO-SGD with a learning rate $\gamma = 1 \times 10^{-8}$ and a batch size of $\sqrt{n}$. For ZO-SPIDER-coord and ZO-SPIDER-rand, the learning rates are set to $\gamma = 10^{-6}$ and $\gamma = 10^{-8}$, respectively. Both methods use an epoch size of $q = \sqrt{n}$, with batch sizes $B = n$ and $B' = \sqrt{n}$. For the proposed PF-VRZO method, we also set the epoch size to $q = \sqrt{n}$, and choose $B = n$ and $B' = \sqrt{n}$ for both the coord and random-direction estimators. *We remark that the setting $q = n$, $B = n$, and $B' = 1$ is also valid, although it yields slightly worse empirical performance in this experiment.*

## E.3 A SMALL EXPERIMENT TO VERIFY THE EFFECTIVENESS OF THE ADAPTIVE SMOOTHING PARAMETER

This is a small experiment designed to demonstrate the effectiveness of our adaptive smoothing parameter. We conducted an ablation experiment (placed at the end of the appendix due to page limits) based on the Nonconvex Phase Retrieval setup in the main text. We compare the following four variants: 1. Original ZO-SPIDER, using step size $\gamma = 0.001$ and $\mu = 1$. 2. ZO-SPIDER-adastep, adaptive step size but fixed $\mu = 1$. 3. ZO-SPIDER-adastep, adaptive step size but fixed $\mu = 0.5$. 4. Our parameter-free PF-VRZO (adaptive step size + adaptive $\mu_t$).

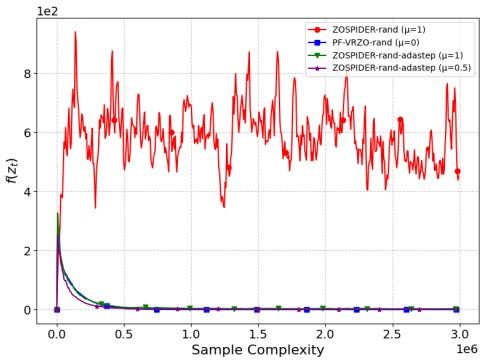 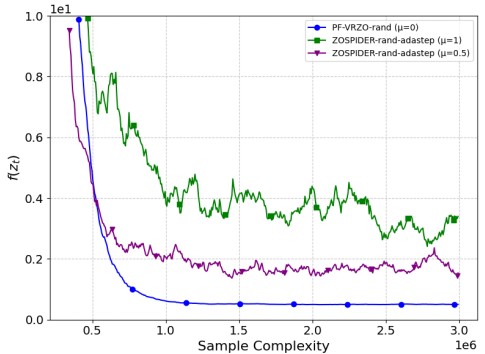

(a) Comparison of four algorithms: The original ZO-SPIDER (red curve) exhibits severe divergence (function value exceeds 600), which obscures the performance of the other three algorithms (with smaller function values).

(b) Zoomed view of the region where $f(z) < 10$ in (a): This magnification clarifies the convergence behaviors of the three algorithms with smaller function values, while our PF-VRZO (blue curve) achieves full optimization.

Figure 4: The original ZO-SPIDER (Group 1) diverges drastically under this parameter setting, with function values surging beyond 600. - Groups 2 and 3 (ZO-SPIDER-adastep) outperform Group 1, yet their function values stagnate (plateauing around 4 and 2, respectively) and fail to decrease further. This aligns with our theoretical analysis: since the fixed $\mu$ does not diminish with $T$, estimator noise accumulates to a point that halts progress. - The $\mu = 0.5$ variant plateaus later than $\mu = 1$—a result consistent with the observation that a smaller fixed $\mu$ delays (but does not resolve) the stagnation issue.Our PF-VRZO (Group 4), which employs an adaptive $\mu_t$, achieves complete optimization successfully.

From the experimental results, we highlight: 1. Adaptive step sizes generally improve convergence behavior. 2. Our adaptive smoothing parameter $\mu_t$ works synergistically with adaptive step sizes. From our theoretical analysis, a fixed $\mu$ cannot shrink as $T$ grows, so the zeroth-order estimator noise eventually fails to meet the increasingly stringent accuracy requirement in later stages of training, causing the algorithm to stall. In contrast, our adaptive $\mu_t$ avoids this issue by design and ensures stable convergence.

$$f(\bar{x}_t) - f(x_*) \leq \frac{1}{\sum_{k=0}^{t-1} \bar{r}_k} \sum_{k=0}^{t-1} \bar{r}_k (f(x_k) - f(x_*)).$$

