# OpenReview forum: "Parameter-Free Variance Reduced Zeroth-Order Optimization for Non-Convex Problems"
_ICLR.cc/2026/Conference — Submitted to ICLR 2026_

### Official Review · Reviewer_WWDX · 2025-10-30

**Soundness:** 3
**Presentation:** 3
**Contribution:** 3
**Rating:** 4
**Confidence:** 3

**Summary:**

The paper proposes PF-VRZO, a parameter-free, variance-reduced zeroth-order framework for nonconvex finite-sum problems. It gives two variants (coordinate-wise and random-direction) that combine SPIDER-type variance reduction with adaptive step size $\gamma_t$ and adaptive smoothing $\mu_t$.

**Strengths:**

Precise wording and clear organization. The paper defines notation early, states assumptions explicitly, and keeps terminology consistent across sections. Algorithms, lemmas, and theorems are numbered and referenced cleanly, which makes the logical flow easy to follow.

Motivation is well articulated. The authors clearly diagnose the tuning sensitivity of zeroth-order methods and motivate combining variance reduction with adaptive step size and smoothing. The problem setting, design goals, and why the proposed choices address concrete pain points are laid out without ambiguity.

**Weaknesses:**

**Question 1**  It would help to state clearly what "parameter-free" means here.  The coordinate variant incurs $O(d)$ function calls.

**Question 2**  Positioning relative to prior work on reducing dimension constrain. Some ZO methods mitigate dimension effects. A brief note on what this work does better-overall query complexity under comparable assumptions, robustness/constant factors, or ease of use-would help readers understand the advantage.

**Question 3**  Distinctiveness beyond ZO-SPIDER. Since the recursion resembles SPIDER, it would be useful to highlight the distinctive elements and show they matter-perhaps with a small ablation against a tuned ZO-SPIDER under the same query budget and a short proof pointer indicating where standard SPIDER would fail without your $\mu_t$ design.

**Question 4**  When should practitioners choose coordinate or random directions?

**Questions:**

And some minor issues:
1. Missing spaces after punctuation, e.g., “dimension d,without relying…” → “dimension d, without relying…”
2. Mixed notation of “non-convex” and “nonconvex”; use one form consistently (preferably nonconvex).
3. In figure captions: “Improperly tuned Parameters” → “Improperly tuned parameters.”
4. References to algorithms (e.g., “ZO-SGD,PF-VRZO,ZO-SPIDER”) lack spacing; should be “ZO-SGD, PF-VRZO, ZO-SPIDER.”

---

> ### Author Response · Authors · 2025-11-25
> **Thank you for taking the time to review our paper. Regarding the questions you raised about our work, we provide detailed responses below.**
>
> **Q1: It would help to state clearly what "parameter-free" means here. The coordinate variant incurs  function calls.**
>
> Thank you for the comment and for giving us the opportunity to clarify the potential confusion. By “param-free,” we mean that the method does not involve any parameters that require repeated trial-and-error tuning or multiple runs of the algorithm. The method uses only quantities such as the dataset size $n$and the model dimension $d$, which are inherent to the problem formulation.
>
> **Q2: Positioning relative to prior work on reducing dimension constrain. Some ZO methods mitigate dimension effects. A brief note on what this work does better-overall query complexity under comparable assumptions, robustness/constant factors, or ease of use-would help readers understand the advantage.**
>
> We thank the reviewer for the insightful question regarding our positioning relative to prior work on reducing the dimension dependence in zeroth-order optimization.
>
> **(1) On reducing the dependence on the ambient dimension.**
> We first clarify that achieving better-than-linear dependence on the dimension d typically requires additional structural assumptions on the objective, such as low kappa-effective rank [1] or newly proposed complexity measures such as $ED     _ {\alpha}$[2]. Under the standard smoothness assumptions adopted in our work, such improvements are impossible: even for the simpler convex setting, [3] shows that any zeroth-order method must incur at least $\mathcal{O}(d)$oracle complexity to obtain an $\epsilon$-stationary point.
> Therefore, under standard assumptions, the dependence on d achieved in our work is already optimal. That said, we agree that reducing the dimension dependence is an interesting and valuable direction. Combining parameter-free zeroth-order methods with such techniques is an appealing future research avenue, and we believe our framework provides a natural starting point for this integration.
>
> **(2) Positioning and advantages relative to existing work.**
> Our contribution is orthogonal to dimensionality-reduction techniques. To the best of our knowledge, this work provides the first parameter-free zeroth-order algorithm for nonconvex finite-sum optimization under standard assumptions, which greatly improves ease of use. Our method does not require any manual tuning of hyperparameters, avoiding multiple runs or grid search.
> In contrast, existing works that reduce the dimension dependence still require tuning at least two key parameters, such as the stepsize and the smoothing parameter. Thus, our method offers a complementary advantage in practical robustness and usability while maintaining optimal dependence on $d$under comparable assumptions.

---

> ### Author Response · Authors · 2025-11-25
>
> **Q3: Distinctiveness beyond ZO-SPIDER. Since the recursion resembles SPIDER, it would be useful to highlight the distinctive elements and show they matter-perhaps with a small ablation against a tuned ZO-SPIDER under the same query budget and a short proof pointer indicating where standard SPIDER would fail without your  design.**
> First, we note that our algorithm differs from ZO-SPIDER in the following structural aspects:
>
> 1) Param-free step size.
> 2) Automatically adjusted smoothing parameter $\mu     _ t$.
> 3) A variance-reduction estimator $v     _ t$ adapted to the adaptive parameter.
>
> When $t$ is not a multiple of $n$, our update is
> $$v     _ t \gets \hat{\nabla}     _ {\mu     _ t} f     _ {i     _ t}(x     _ t) - \hat{\nabla}     _ {\mu     _ {t-1}} f     _ {i     _ t}(x     _ {t-1}) + v     _ {t-1},$$
> whereas in ZO-SPIDER the update is
> $$v     _ t \gets \hat{\nabla}     _ {\mu} f     _ {i     _ t}(x     _ t)- \hat{\nabla}     _ {\mu} f     _ {i     _ t}(x     _ {t-1})+ v     _ {t-1}.$$
> Note that we deliberately use different smoothing parameters $\mu     _ t$ and $\mu     _ {t-1}$ for the zeroth-order estimators at $x     _ t$ and $x     _ {t-1}$.
>
> ---
> Then, we point out that under adaptive step sizes, using a fixed smoothing parameter can fail. As we state in the paper:
> Based on our results, after $T$ iterations the accumulated error is approximately
> $$\mathcal{O} \left[\frac{1}{T} \left(\sum     _ {t=0}^{T-1}\mu     _ t^2 + \sum     _ {t=0}^{T-1}\mu     _ t\right)+ \frac{1}{\sqrt{T}} \left(n^{\frac{5}{4}}\sum     _ {t=0}^{T-1}\mu     _ t^{2}+ n^{\frac{1}{2}}\sqrt{\sum     _ {t=0}^{T-1} 2\mu     _ t^{2}}\right)\right].$$
>
> To ensure convergence, it is crucial that
> $$\sum     _ {t=0}^{T-1} \mu     _ t \le \mathcal{O}(\sqrt{T}), \qquad \sum     _ {t=0}^{T-1} \mu     _ t^2 \le \mathcal{O}(1).$$
>
> Otherwise, the algorithm may diverge. This shows that the error grows with $T$. A natural idea is to let the smoothing parameter $\mu$ depend on $T$, which would automatically guarantee these two conditions. However, this conflicts with the goal of designing a parameter-free algorithm, since $T$ is not known beforehand.
>
> Using a fixed $\mu$ may cause an awkward situation: a value of $\mu$ tuned to work well in early iterations may become too large as $T$ grows, precisely when convergence requires increasingly smaller $\mu$. This leads to the algorithm getting stuck at a certain accuracy level and failing to further optimize. Our adaptive smoothing parameter $\mu     _ t$ avoids this issue, and we will verify this through experiments.
>
> ---
>
> According to your suggestion, we conducted an ablation experiment (placed at the end of the appendix due to page limits) based on the Nonconvex Phase Retrieval setup in the main text. We compare the following four variants:
> 1. Original ZO-SPIDER, using step size $\gamma=0.001$ and $\mu=1$.
> 2. ZO-SPIDER-adastep, adaptive step size but fixed $\mu=1$.
> 3. ZO-SPIDER-adastep, adaptive step size but fixed $\mu=0.5$.
> 4. Our parameter-free PF-VRZO (adaptive step size + adaptive $\mu     _ t$).
>
> **Experimental observations:**
>
> - The original ZO-SPIDER (Group 1) diverges severely under this parameter setting, with function values blowing up beyond 600.
> - Groups 2 and 3 (ZO-SPIDER-adastep) perform better than Group 1, but the function values plateau at around 4 and 2, respectively, and cannot decrease further. This matches our theoretical argument: because $\mu$ cannot decrease with $T$, the estimator noise eventually becomes too large for further progress.
> - The version with $\mu=0.5$ plateaus later than $\mu=1$, consistent with the fact that a smaller fixed $\mu$ delays—but does not solve—the issue.
> - Our PF-VRZO (Group 4), equipped with adaptive $\mu     _ t$, successfully completes the optimization.
> ---
>
> **From the experimental results, we highlight:**
> 1. Adaptive step sizes generally improve convergence behavior.
> 2. Our adaptive smoothing parameter $\mu     _ t$ works synergistically with adaptive step sizes. From our theoretical analysis, a fixed $\mu$ cannot shrink as $T$ grows, so the zeroth-order estimator noise eventually fails to meet the increasingly stringent accuracy requirement in later stages of training, causing the algorithm to stall. In contrast, our adaptive $\mu     _ t$ avoids this issue by design and ensures stable convergence.

---

> ### Author Response · Authors · 2025-11-25
>
> **Q4: When should practitioners choose coordinate or random directions?**
>
> Since our theoretical analysis shows that the two methods have the same order of complexity, we consider this to be a relatively open question. Our perspective on this is as follows:
> For the coordinate-wise method, the absence of randomness provides better stability. Additionally, since there is no need to generate random numbers (which in practice requires some computational effort), it saves one step during code execution.
> For the random-directions method, we see two advantages. First, based on studies on escaping saddle points (e.g., [4]), introducing some random noise into the algorithm can help it escape saddle point and increase the probability of reaching the  optimum. Second, as the reviewer noted, for problems with certain structures, the number of iterations $T$ required by the random-directions method may be significantly smaller than the worst-case estimate given by the standard analysis.
>
> **REFERENCES:**
>
> [1]DPZero: Private FineTuning of Language Models without Backpropagation
>
> [2]Zeroth-order Optimization with Weak Dimension Dependency
>
> [3]Optimal rates for zero-order convex optimization: the power of two function evaluations
>
> [4]How to escape saddle points efficiently.

---

### Official Review · Reviewer_aj3g · 2025-10-31

**Soundness:** 3
**Presentation:** 1
**Contribution:** 2
**Rating:** 4
**Confidence:** 4

**Summary:**

This paper introduces PF-VRZO (Parameter-Free Variance Reduced Zeroth-Order), a parameter-free optimization method for non-convex and finite-sum problems in black-box settings. The proposed method obtains the optimal query complexity for non-convex optimization problem.

**Strengths:**

The proof sketch is presented clearly. It builds upon the framework of the previous work SPIDER and effectively incorporates the properties of the zeroth-order optimization method.

**Weaknesses:**

1. The proof process is somewhat difficult to follow. For instance, in the proof of Lemma B.2 and Line 379, it is unclear how the authors derive the bound for $x_s - x_{s-1}$. In addition, there are several typos. For example, in Line 906, the author should check the exponent of T in the second term; the definition of $\Phi(T)$ given below is inconsistent with that in Line 900—the second term should be $\mathcal{O}(n^2)$ as stated in the proof.
The authors should carefully check the proof and correct typos to enhance readability.

2. The experimental evaluation is not sufficient. The input data dimensions in both datasets are relatively small. I suggest that the authors include additional experiments on larger or more diverse datasets, to better demonstrate the scalability and generalization of the proposed method.

**Questions:**

Since the proof is not entirely clear to me, my main concerns are related to the theoretical analysis:
- In the proof of Lemma B.2 and Line 379, it is unclear how the authors derive the bound for $x_s - x_{s-1}$;
- It is also not clear how Lemma A.7 is applied to obtain the inequality in Line 942;
- how does the parameter learning rate affect the proof analysis

In Line 63, the authors demonstrate that 'As acknowledged by the authors, extending this result to the nonconvex setting is nontrivial'. Could the authors elaborate on how their proof sketch differs from spider, and clarify which part of the extension to the nonconvex setting is nontrivial?

---

> ### Author Response · Authors · 2025-11-25
> **Thank you for taking the time to review our paper. Regarding the questions you raised about our work, we provide detailed responses below.**
>
> **W1: There are some typos related to $\Phi(T)$**
>
> We sincerely apologize for the typos in the parts related to $\Phi (T)$. You are absolutely right that the exponent of  $T$ in the second term was incorrect in Line 906, and the definition of $\Phi (T) $was inconsistent with Line 900. We have carefully corrected all related typos in the revised version. We thank the reviewer for pointing this out and helping us improve the clarity of the paper.
>
> **W2: The experimental evaluation is not sufficient. The input data dimensions in both datasets are relatively small. I suggest that the authors include additional experiments on larger or more diverse datasets, to better demonstrate the scalability and generalization of the proposed method.**
>
> To address the concern about insufficient experimental evaluation, we have expanded the DRO experiments: originally limited to 1 dataset (Life Expectancy), we now include **3 datasets in total** (adding Arcene and Communities and Crime, as shown in Figure 3). These datasets vary significantly in input feature dimensions: Life Expectancy has 19 features, Communities and Crime has 122 features and Arcene has 10,000 features. This expanded setup (covering small, medium, and large feature scales) better demonstrates the scalability and generalization of the proposed PF-VRZO method across diverse DRO task scenarios.
>
> **Q1: In the proof of Lemma B.2 and Line 379, it is unclear how the authors derive the bound for $x _{s}-x _{s-1}$**
>
> At this step, using the update rule of the algorithm, we have
> $\lVert x _{s}-x _{s-1} \rVert=\frac{1}{n^{1/4}c}* \underbrace{\lVert \frac{v _{t}}{\sqrt{\left( n^{1/2} + \sum _{s=0}^t \|v _{s}\|^2 \right)}} \rVert } _{\leq 1}\leq \frac{1}{n^{1/4}c}$. We have added this explanation to the paper to make the derivation clearer.
>
> **Q2: It is also not clear how Lemma A.7 is applied to obtain the inequality in Line 942**
>
> According to the iterative format of the algorithm, we have $$\sum _{t=0}^{T-1} \gamma _t^2 \cdot \| v _t \|^2 = \frac{1}{c^2 \sqrt{n}} \cdot \sum _{t=0}^{T-1} \frac{\| v _t \|^2}{\sqrt{n} + \sum _{s=0}^t \| v _s \|^2}.$$Then, by applying Lemma A.7 $$\sum _{\ell=1}^T \frac{a _\ell}{1 + \sum _{i=1}^\ell a _i} \leq \log\left(\sum _{i=1}^T a _i + 1\right)$$, we can obtain $\sum _{t=0}^{T-1} \frac{\| v _t \|^2}{\sqrt{n} + \sum _{s=0}^t \| v _s \|^2} \leq \sum _{t=0}^{T-1} \frac{\| v _t \|^2}{1 + \sum _{s=0}^t \| v _s \|^2} \leq \log\left( \sum^{T-1} _{t=0} \| v _{t} \|^2 \right) \leq \log (\Phi(T))$Combining the above results, we can get $$\sum _{t=0}^{T-1} \gamma _t^2 \cdot \| v _t \|^2 \leq \frac{1}{c^2 \sqrt{n}} \log(\Phi(T)).$$
> **Q3: how does the parameter learning rate affect the proof analysis**
>
> My understanding is that the reviewer is asking how the adaptive step size affects our convergence analysis. Since the step size is time-varying and depends on both the gradient and the iteration counter, it introduces several additional challenges in the analysis. Below we highlight three representative examples.
> (1) Finding an upper bound for terms related to the gradient estimator $\sum _{t=0}^{T-1} \mathbb{E}\big[\gamma _t \|v _t\|^2\big]$.
> (2) Bounding the zeroth-order–related error term $\sum _{t=0}^{T-1} \gamma _t \delta _t^2$.
> (3) Establishing a recursive relation for $\mathbb{E}\big[\gamma _t \|v _t - \bar{\nabla} _{\mu _t} f (x _t)\|^2 \mid \mathcal{F} _{t-1}\big]$ when applying SPIDER.
> To address these issues, we rely on three key properties of the adaptive step size:
> 1. **Relationship between step size and accumulated gradients.**
>    Using the relation between $\gamma _t$and the cumulative squared gradients, we derive the logarithmic bound
> $$\sum _{t=0}^{T-1} \frac{\|v _t\|^2}{\sqrt{n} + \sum _{s=0}^t \|v _s\|^2} \le\sum _{t=0}^{T-1} \frac{\|v _t\|^2}{1 + \sum _{s=0}^t \|v _s\|^2} \le \log\!\left(\sum _{t=0}^{T-1} \|v _t\|^2\right) \le \log(\Phi(T)),
>    $$
>    which controls terms related to $\sum _{t=0}^{T-1} \mathbb{E}\big[\gamma _t \|v _t\|^2\big]$.
> 2. **Boundedness of the step size.**
>    Since $\gamma _t \le 1/(c n^{1/4})$, we can factor out the step-size term from the zeroth-order error term $\sum _{t=0}^{T-1} \gamma _t \delta _t^2$and bound it accordingly.
> 3. **Monotonicity of the step size.**
>    Because $\gamma _t$is monotonically decreasing, we have
> $$\mathbb{E}\big[\gamma _t \|v _t - \nabla f(x _t)\|^2 \mid \mathcal{F} _{t-1}\big]\le\mathbb{E}\big[\gamma _{t-1} \|v _t - \nabla f(x _t)\|^2 \mid \mathcal{F} _{t-1}\big].$$
>    Moreover, $\gamma _{t-1}$ is independent of the error term $\|v _t - \nabla f(x _t)\|^2$ , so
> $$\mathbb{E}\big[\gamma _{t-1} \|v _t - \nabla f(x _t)\|^2 \mid \mathcal{F} _{t-1}\big] = \gamma _{t-1} \cdot \mathbb{E}\big[\|v _t - \nabla f(x _t)\|^2 \mid \mathcal{F} _{t-1}\big]. $$
>    This separates the step-size factor from the remaining error term.
> These three properties together allow us to control all step-size–related quantities in the convergence proof.

---

> ### Author Response · Authors · 2025-11-25
>
> **Q4: In Line 63, the authors demonstrate that 'As acknowledged by the authors, extending this result to the nonconvex setting is nontrivial'. Could the authors elaborate on how their proof sketch differs from spider, and clarify which part of the extension to the nonconvex setting is nontrivial?**
>
> **For POEM**, in order to convert the cumulative objective into several parts that are easy to analyze, it is essential to use Jensen’s inequality based on convexity, namely
>
> $$f(\bar{x}    _ t) - f(x    _ *) \le \frac{1}{\sum    _ {k=0}^{t-1} \bar{r}    _ k} \sum    _ {k=0}^{t-1} \bar{r}    _ k \big( f(x    _ k) - f(x    _ *) \big).$$
> Then, using the convexity of $f(x)$, we apply the convexity of $f     _ {\mu     _ k}(x)$.
> Using the standard property of convex functions
> $$f(w)\le f(w') + \nabla f(w')^\top (w - w') + \frac{L}{2}\|w - w'\|^2,$$
> we obtain
> $$f(\bar{x}     _ t) - f(x     _ *)\le\frac{1}{\sum     _ {k=0}^{t-1}\bar{r}     _ k}\sum     _ {k=0}^{t-1}\bar{r}     _ k ( f     _ {\mu     _ k}(x     _ k) - f     _ {\mu     _ k}(x     _ *) + 2L\mu     _ k \big),$$
> and further
> $$f(\bar{x}     _ t) - f(x     _ *)\le\frac{1}{\sum     _ {k=0}^{t-1}\bar{r}     _ k}\sum     _ {k=0}^{t-1}\bar{r}     _ k \big( \langle \nabla f     _ {\mu     _ k}(x     _ k), x     _ k - x     _ * \rangle + 2L\mu     _ k \big).$$
> Then split the sum above into the following three parts:
> $$\sum     _ {k=0}^{t-1} \bar{r}     _ k \langle g     _ k, x     _ k - x     _ * \rangle+\sum     _ {k=0}^{t-1} \bar{r}     _ k \langle \Delta     _ k, x     _ k - x     _ * \rangle+\sum     _ {k=0}^{t-1} 2L \bar{r}     _ k \mu     _ k.$$
> **In contrast**, we do not rely on the convexity of the objective. Instead, we directly split the cumulative objective into three parts.
> The main difficulty lies in the careful rearrangement of these terms and in adapting SPIDER such that the coupling between adaptive steps and the adaptive smoothing parameter can be established:
> $$\begin{align}&\frac{1}{T}\sum     _ {t=1}^T \|\nabla f(x     _ t)\|\le \frac{1}{T} \sum     _ {t=1}^T \mathbb{E}\|v     _ t\|+
> \frac{1}{T}\sum     _ {t=1}^T \mathbb{E}\|v     _ t - \hat{\nabla}     _ {\mu     _ t} f(x     _ t)\|+\frac{1}{T}\sum     _ {t=1}^T \|\hat{\nabla}     _ {\mu     _ t} f(x     _ t) - \nabla f(x     _ t)\|.\end{align}$$
> In summary, POEM cannot be extended to the nonconvex setting mainly because its analysis relies crucially on convexity.
> Convexity allows the cumulative objective to be decomposed into four independent subproblems, each of which can be analyzed separately.
> Although we also decompose the cumulative objective into four analyzable parts, our transition does not rely on convexity, and the key challenge is the coupling among these terms in the nonconvex setting.

---

### Official Review · Reviewer_fY9W · 2025-11-01

**Soundness:** 3
**Presentation:** 3
**Contribution:** 3
**Rating:** 4
**Confidence:** 3

**Summary:**

In this paper, the authors studied parameter-free algorithms for stochastic nonconvex optimization in the zeroth-order setting. They proposed PF-VRZO, a parameter-free variance-reduced zeroth-order optimization method, which requires only the problem dimension and sample size as inputs. The method adaptively adjusts both the smoothing parameter and the step size, eliminating the need for manual hyperparameter tuning. It introduces two algorithmic variants, leveraging coordination-wise and random-direction estimators respectively. The authors prove non-asymptotic convergence guarantees, showing PF-VRZO reaches a function-query complexity of $\tilde{\mathcal{O}}(d\sqrt{n}\epsilon^{-2})$. They further validate the method empirically on non-convex phase retrieval and distributionally robust optimization.

**Strengths:**

- The idea of parameter-free zeroth-order nonconvex optimization is interesting and novel.
- The method removes the need for tuning smoothing and step-size parameters, which is an advantage in practice.

**Weaknesses:**

- The paper is not well written. For example, Section 3.5 mostly consists of lemmas / theorems, but lacks explanation about their significance and connection. Also, there are many typos / formatting issues (please see below).
- The experiment results are not very promising. The proposed method does not consistently outperform existing methods across different datasets and metrics.

**Questions:**

- Questions
  - Section 2 is very compact. Is it necessary to put the content in a separate section?
  - In Remark 1 and 3, how to derive the $\mathcal{O}(1+ n/n)$ complexity?
  - In the experiments, the random variant performs better than the coordinate variant in terms of sample complexity. But why does the coordinate variant perform better in terms of running time?
- Suggestions
  - The captions of the figures are short. I suggest the authors provide more explanation about the figure in the caption.
  - It would be better to have some texts explaining Algorithm 1.
  - The palettes of the figures are not consistent. I suggest the authors use the same color for the same method across different figures.
- Typos
  - Line 60: (Ren & Luo, 2025) -> Ren & Luo (2025)
  - Line 185: (Gao et al., 2018) -> Gao et al. (2018)
  - Line 189: unbiased estimator of \hat{\nabla} -> unbiased estimator of \nabla
  - Line 192: random-direction -> Random-direction

---

> ### Author Response · Authors · 2025-11-25
> **Thank you for taking the time to review our paper. We have incorporated content to enhance readability in the revised version of the paper in accordance with your suggestions, and we provide responses to your questions below.**
>
> **W1: Section 3.5 mostly consists of lemmas / theorems, but lacks explanation about their significance and connection.**
>
> Thank you for your valuable suggestion! We have optimized Section 3.5 by supplementing explanations of core significance and illustrations of logical connections after the relevant lemmas/theorems.
>
> **W2: The experiment results are not very promising. The proposed method does not consistently outperform existing methods across different datasets and metrics.**
>
> A: First, we would like to clarify that all existing methods compared in the figures involve one or more parameters that require repeated adjustment. Our tuning-free method is benchmarked against these algorithms after extensive parameter tuning In practice, existing methods often need multiple parameter adjustments to ensure normal convergence, and even more adjustments to achieve favorable performance. The total time and effort consumed by such repeated tuning are far greater than those of our tuning-free method.
> Additionally, we would like to highlight an additional advantage: when bugs occur during code execution or results are unsatisfactory, one cannot help but suspect that the suboptimal performance may stem from inadequate parameter settings of non-tuning-free methods. In contrast, our tuning-free method eliminates this possibility theoretically by design, as it requires no parameter adjustment—this is significantly more elegant for troubleshooting and code implementation.
>
> **Q1: Section 2 is very compact. Is it necessary to put the content in a separate section?**
>
> Following your suggestion, we have revised this section into a short paragraph.
>
> **Q2: In Remark 1 and 3, how to derive the  $\mathcal{O}\left( 1+\frac{n}{n} \right)$ complexity?**
>
> When $t$is an integer multiple of $n$, we consume $n$zeroth-order estimators to compute $v _t$; otherwise, we consume $\mathcal{O}(1)$zeroth-order estimators for this computation. Consequently, a total of $\mathcal{O}(2n)$zeroth-order estimators are consumed over every $n$iterations, which translates to a cost of $\mathcal{O}\left(1 + \frac{n}{n}\right)$zeroth-order estimators  per iteration.
>
> **Q3: In the experiments, the random variant performs better than the coordinate variant in terms of sample complexity. But why does the coordinate variant perform better in terms of running time?**
>
> We believe that this phenomenon is mainly due to the additional computational overhead required by the random estimator.  Although the random variant has better sample complexity, each zeroth-order query requires generating a random direction uniformly distributed on the $d$-dimensional unit sphere.  In our implementation, this is done by first sampling a $d$-dimensional Gaussian vector and then normalizing it, i.e., $u \sim \mathcal{N}(0, I _d),\quad v = \frac{u}{\|u\| _2},$ so that $v$is uniformly distributed on the unit sphere.  Compared to the coordinate-wise estimator, which only perturbs one coordinate at a time and therefore incurs almost no sampling cost, this additional sampling and normalization step introduces noticeable runtime overhead.  As a result, even though the random estimator uses fewer function evaluations, the coordinate-wise method can still achieve faster wall-clock time in practice.
>
> **S1: The captions of the figures are short. I suggest the authors provide more explanation about the figure in the caption.**
>
> Thank you for your suggestion. We have added explanations to the figure captions as you recommended.
>
> **S2: It would be better to have some texts explaining Algorithm 1.**
>
> Thank you for your suggestion. We have added some explanations to Algorithm 1 and Algorithm 2 as you recommended.
>
> **S3: The palettes of the figures are not consistent. I suggest the authors use the same color for the same method across different figures.**
>
> Thank you for your suggestion. Following your advice, we have revised the experimental figures to ensure that the same method is represented by the same color across different figures.

---

### Official Review · Reviewer_rpNE · 2025-11-03

**Soundness:** 3
**Presentation:** 2
**Contribution:** 3
**Rating:** 6
**Confidence:** 3

**Summary:**

This paper studies zero-th order optimization of a smooth non-convex objective that is written as finite sum of n smooth terms.
The proposed algorithm is parameters free and implement variance reduction.
The authors prove that time-averaged norm of the gradient converges below \eps in a number of stochastic steps that is of order at most d*n^{1/2}/\eps^2 (where d is the ambient dimension and n is the number of terms in the optimization.

Recent work on parameters free zeroth order optimization was limited to convex functions (but also proved stronger results, namely convergence of the cost).

**Strengths:**

1. Zeroth order, parameters free optimization is of course a well motivated area of study.
2. The results appear to be novel and sound hence filling a gap in earlier literature.

**Weaknesses:**

1. The algorithm seems a variant over earlier parameter-free papers (Ivgi et al 2023; and follow up work on first order method also cited in the present paper), as combined with smoothing techniques for gradient estimation using zero-th order methods (eg Duchi et al 2015 and follow up work, Ren Luo 2025)

2. Also for what concerns the analysis, the paper seems closely related to proof techniques from earlier literature on zeroth order methods (in particular Ji et al 2019).

3. I understand that proving that the average gradient converges to zero has become the standard form of results in this subcommunity. However, I am still not sure why one should care about this from a machine learning perspective, since the algorithm can converge on a suboptimal minimum.

**Questions:**

The first questions are related to the points mentioned above:

1) Can you please spell out the algorithmic innovation. Is it the specific choice of \mu_t?
If so I would at least explain upfront  where this choice comes from in the analysis.

2) Similarly for the algorithm analysis. Is there a new proof component with respect to earlier proofs?

---

> ### Author Response · Authors · 2025-11-25
> **Thank you for taking the time to review our paper. Regarding the questions you raised about our work, we provide detailed responses below.**
>
> **W1: I understand that proving that the average gradient converges to zero has become the standard form of results in this subcommunity. However, I am still not sure why one should care about this from a machine learning perspective, since the algorithm can converge on a suboptimal minimum.**
>
> We thank the reviewer for raising this conceptual question.
>
> **(1) Global minimization in general nonconvex problems is computationally intractable.**
> As established in classical complexity results (e.g., [2]), finding a global minimum of a  nonconvex function is NP-hard in the worst case. Therefore, the standard objective in modern nonconvex optimization is to find an ε-stationary point rather than a global optimum.
>
> **(2) Convergence of the averaged gradient implies the existence of an ε-stationary iterate.**
> Although our theorem bounds the averaged gradient norm, $\frac{1}{T}\sum _{t=1}^T \mathbb{E}\|\nabla f (x _t)\|^2 \le \varepsilon^2$,
> this directly ensures that at least one iterate satisfies  $\|\nabla f (x _{t^\ast})\| \le \varepsilon.$
> This is the standard and widely accepted convergence criterion in nonconvex zeroth-order and first-order optimization. In practice, one simply outputs the best iterate.
>
> **(3) For many machine learning models, local minima are already meaningful or even globally optimal.**
> As summarized in [3], a large body of work has shown that, for several important ML problems, every local minimum is a global minimum, including tensor decomposition, dictionary learning, matrix sensing, matrix completion, and phase retrieval (e.g., Ge et al. 2015; Sun et al. 2016a, b; Bhojanapalli et al. 2016). Certain classes of deep neural networks also exhibit this property (Kawaguchi 2016). Moreover, empirical and theoretical evidence suggests that, in more general deep networks, most local minima are already close to globally optimal (Choromanska et al. 2014).
> For these reasons, the guarantee of finding an ε-stationary point is both theoretically justified and practically relevant in machine learning.

---

> ### Author Response · Authors · 2025-11-25
>
> **Q1: Can you please spell out the algorithmic innovation? Is it the specific choice of $\mu _t$? If so I would at least explain upfront where this choice comes from in the analysis.**
>
> Thank you for your question. In fact, the differences are not limited to μₜ. In terms of algorithmic design, our method differs from ZO-SPIDER (Ji et al 2019) [1] in four key aspects:
>
> **(1) Pram-free step size.**
>    We adopt a parameter-free adaptive step size, whereas [1] uses a fixed step size that needs to be manually tuned.
>
> **(2) Automatically adjusted smoothing parameter $\mu _t$.**
>    We use a parameter-free adaptive smoothing parameter $\mu _t$. In contrast, the choice of $\mu$ in [1] requires prior knowledge of the total number of iterations $T$ needed for convergence. This is impractical in real applications, since one cannot know the required number of iterations before running the algorithm.
>
> **(3)The variance reduction estimator $v _{t}$improved to adapt to the adaptive parameter.**
>    When $t$ is not a multiple of $n$, our update is
>    $$v _t \gets \hat{\nabla} _{\mu _t} f _{i _t}(x _t) - \hat{\nabla} _{\mu _{t-1}} f _{i _t}(x _{t-1}) + v _{t-1},$$
>    whereas in [1] the update is
>    $$v _t \gets \hat{\nabla} _{\mu} f _{i _t}(x _t) - \hat{\nabla} _{\mu} f _{i _t}(x _{t-1}) + v _{t-1}.$$
>    Note that we deliberately use different  smoothing parameters $\mu _t$ and $\mu _{t-1}$ for the zeroth-order estimators at $x _t$ and $x _{t-1}$. This modification is crucial because introducing adaptive, non-constant step sizes and smoothing parameters creates additional difficulties in the analysis, and this adjustment is necessary to guarantee our convergence results.
>
> **(4)Including a stochastic zeroth-order estimator version with distinct characteristics**
>
> Paper[1] proposes only one algorithm, ZO-SPIDER-Coord, which employs a coordinate-wise estimator for the SPIDER algorithm, while we additionally introduce PF-VRZO (Random-direction).
>    ZO-SPIDER-Coord requires $\mathcal{O}(d)$ function evaluations per iteration to compute the SPIDER estimator $v _t$, while PF-VRZO (Random-direction) only needs $\mathcal{O}(1)$. Of course, PF-VRZO requires $\mathcal{O}(d)$ more iterations, so the two methods end up having the same total number of function evaluations. Even though the overall complexity matches [1], the decomposition of the complexity is different, which we believe is an interesting distinction.
>
> **REMARK**: We would like to remind you that Section 1 of our paper includes a Q1/A1 and Q2/A2 discussion, where we elaborate on the motivation behind these design choices.

---

> ### Author Response · Authors · 2025-11-25
>
> **Q2: Similarly for the algorithm analysis. Is there a new proof component with respect to earlier proofs?**
>
> Thank you for the question. Indeed, our analysis requires several new proof components compared with earlier works. The main difficulty comes from combining a time-varying smoothing parameter $\mu _t$with an adaptive step size. This interaction creates challenges not present in analyses that use fixed parameters.
> A first challenge is to control the accumulation of the zeroth-order estimation error throughout the optimization process. Our analysis shows that after $T$iterations, the accumulated error scales as
>
> $$\mathcal{O} \left[\frac{1}{T} \left(\sum     _ {t=0}^{T-1}\mu     _ t^2 + \sum     _ {t=0}^{T-1}\mu     _ t\right)+ \frac{1}{\sqrt{T}} \left(n^{\frac{5}{4}}\sum     _ {t=0}^{T-1}\mu     _ t^{2}+ n^{\frac{1}{2}}\sqrt{\sum     _ {t=0}^{T-1} 2\mu     _ t^{2}}\right)\right].$$
> To ensure convergence, it is necessary that
> $$\sum _{t=0}^{T-1}\mu _t \le \mathcal{O}(\sqrt{T}), \qquad\sum _{t=0}^{T-1}\mu _t^2 \le \mathcal{O}(1).$$
> One might attempt to choose $\mu _t$directly based on $T$so that these conditions hold, but this contradicts our goal of designing a parameter-free method, since $T$is not known in advance. To overcome this issue, we introduce the automatically decaying sequence
> $$\mu _t = \frac{1}{(t+1)\sqrt{nd}},$$
> which guarantees the required bounds without any manual tuning. This design and its analysis are new compared with prior zeroth-order methods.
> A second challenge concerns the compatibility between a time-varying $\mu _t$and the recursive structure required by variance-reduction methods such as SPIDER. The standard recursion
>
> $$\mathbb{E}\|v _t - \bar{\nabla} _{\mu _t} f(x _t)\|^2\le\mathbb{E}\|v _{t-1} - \bar{\nabla} _{\mu _{t-1}} f(x _{t-1})\|^2+ \text{(additional terms)}$$
>
> holds only if the following identity is preserved:
>
> $$\mathbb{E}\!\left[\bar{\nabla} _{\mu _1} f _{i _t}(x _t)-\bar{\nabla} _{\mu _2} f _{i _t}(x _{t-1})\right]=\bar{\nabla} _{\mu _t} f(x _t)-\bar{\nabla} _{\mu _{t-1}} f(x _{t-1}).$$
>
> Simply using $\mu _1 = \mu _2 = \mu _t$breaks this equality and destroys the recursion. A key technical component of our proof is therefore to use a slightly asymmetric choice:
> $$\mu _1 = \frac{1}{(t+1)\sqrt{nd}}, \qquad\mu _2 = \frac{1}{t\sqrt{nd}},$$
> which restores the required recursion and allows variance reduction to work seamlessly with an adaptive, parameter-free zeroth-order estimator.
> These proof components lead to concrete differences in the analyses of Lemma C.4 (part I (2)), Theorem C.1, Lemma D.5 (part I (2)), and Theorem D.1 in the latest version of the paper.
>
>
> **REFERENCES:**
>
> [1]Kaiyi Ji, Zhe Wang, Yi Zhou, and Yingbin Liang. Improved zeroth-order variance reduced algorithms and analysis for nonconvex optimization. In International Conference on Machine Learning, 2019.
>
> [2]Murty, K.G., Kabadi, S.N. Some NP-complete problems in quadratic and nonlinear programming. Mathematical Programming 39, 117–129 (1987). https://doi.org/10.1007/BF02592948
>
> [3]Chi Jin, Rong Ge, Praneeth Netrapalli, Sham M Kakade, and Michael I Jordan. How to escape saddle points efficiently. In International Conference on Machine Learning, pages 1724–1732. PMLR, 2017.

---

### Meta-Review · Area_Chair_qXnP · 2026-01-19

**Summary:**

This paper proposes PF-VRZO, a parameter-free, variance-reduced zeroth-order optimization method for nonconvex finite-sum problems. The authors provide theoretical convergence guarantees and validate the method empirically on nonconvex phase retrieval and distributionally robust optimization tasks.

- Reviewer rpNE has concerns about the algorithmic novelty relative to prior work. Specifically, the proposed algorithm appears largely a variant of earlier parameter-free methods (e.g., Ivgi et al., 2023) combined with standard smoothing techniques for zeroth-order gradient estimation (e.g., Duchi et al., 2015; Ren & Luo, 2025).

- Reviewer fY9W finds that experiments do not consistently outperform existing methods.

- Reviewer aj3g has concerns about unclear or difficult-to-follow proofs, typos, and limited experimental evaluation on small datasets.

- Reviewer WWDX requests a clearer explanation of what “parameter-free” means and distinctiveness from ZO-SPIDER.

**Reviewer Concerns:**

Overall, the authors addressed most reviewer concerns with detailed explanations and revisions. However, they have not resolved Reviewer rpNE’s key concern about the algorithmic novelty relative to prior work.

**Reviewer Scores:**

Reviewer rpNE would likely maintain or lower their score, as the key concern regarding the algorithmic novelty relative to prior work remains unresolved.

---

### Decision · Program_Chairs · 2026-01-26

Reject